# ROBUST SEMANTIC SAMPLE FILTERING FOR PARTIALLY VIEW-ALIGNED CLUSTERING

## ABSTRACT

Multi-view Clustering (MvC) typically assumes strict sample alignment across views. However, this assumption often fails to hold in real-world scenarios due to data acquisition or occlusions, resulting in partially view-aligned data. Existing methods tend to rely on prior alignment knowledge and discard unaligned samples during training, hindering their performance and practical applicability. To address this, we propose a novel framework named REFINE that integrates Cross-view Semantics-based Filtering and Shared-space Contrastive Learning to robustly handle partially view-aligned data. Our method dynamically identifies reliable samples by aligning pseudo-labels across views and filters out noisy correspondences to improve clustering prototype initialization and cross-view consistency learning. Moreover, we employ a cross-view decoder to project features into a shared latent space, bridging modality gaps and facilitating more effective contrastive learning. Extensive experiments across five benchmark datasets under both fully aligned and partially aligned settings demonstrate that our approach achieves state-of-the-art performance, delivering superior robustness and generalization in real-world scenarios without strict alignment requirements. Our code has been made anonymously available at `https://github.com/REFINE-REFINE/REFINE`.

## 1 INTRODUCTION

Multi-view Clustering (MvC) aims to leverage the consistency and complementarity across different views to enhance clustering performance (Lebeau et al., 2024; Eisenberg et al., 2025). Most existing MvC methods rely on the strong assumption of perfectly aligned views (Huang et al., 2023; Long et al., 2025). However, this assumption often fails in real-world applications due to issues like asynchronous sensor acquisition, occlusions, or transmission errors, resulting in partially misaligned multi-view data (Zhao et al., 2025; Wang et al., 2024a).

To address this practical challenge, Partially View-aligned Clustering (PVC) has emerged as a promising solution for handling view misalignment. As shown in Figure 1 (a), unlike conventional MvC methods, PVC relaxes strict cross-view correspondence requirements. Recent studies typically assume prior knowledge of the correspondence status, distinguishing between aligned and unaligned samples (Qian et al., 2024; Lu et al., 2024). These approaches leverage the aligned data to extract shared semantic representations and build robust cluster structures, which are then used to correct the correspondences of unaligned samples. However, such reliance on known correspondences limits practical applicability and results in inefficient use of available data, ultimately compromising model generalization. Therefore, a more realistic and challenging scenario arises, where the cross-view correspondences are completely unknown. Only limited works have attempted to address this setting. For instance, CANDY (Guo et al., 2024) leverages high-order semantic graphs with spectral denoising, while ROLL (Sun et al., 2025) employs a weighted cross-entropy loss to mitigate noisy pseudo-labels. However, these methods mainly rely on loss design, which may lead to underfitting of correct correspondences, resulting in suboptimal performance.

To address this weakness, we employ sample filtering, which preserves robustness by discarding unreliable correspondences while fully leveraging trustworthy data. As illustrated in Figure 1 (b), we draw inspiration from Noisy Label Learning (NLL), where it has been observed that deep neural networks tend to capture simple and reliable patterns before memorizing noisy labels (Li et al., 2020).

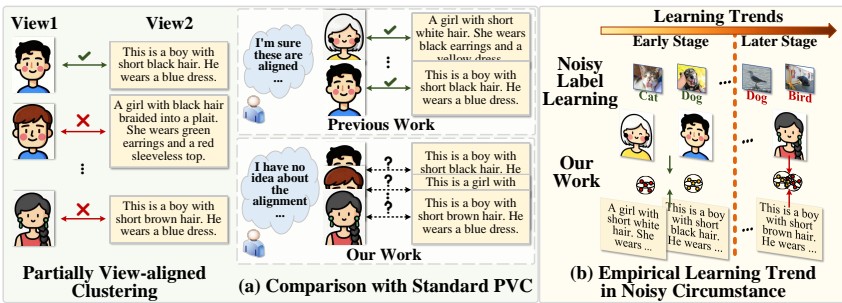

Figure 1: (a) Comparison with Standard PVC: PVC refers to a multi-view clustering scenario where some samples are partially misaligned across views. Existing methods typically rely on a subset of view-aligned data. In contrast, our method addresses the more realistic and challenging setting where alignment information is noisy and completely unknown. (b) Empirical Learning Trend in Noisy Circumstance: Inspired by sample selection strategies in NLL, we observe that models tend to first learn clearer (likely clean) cross-view correspondences before gradually adapting to more complex (potentially noisy) cross-view relationships in partially aligned data.

We argue that the principle can be applied to PVC: semantic consistency across views can serve as a criterion for identifying reliable samples. Specifically, samples with consistent cross-view predictions are more likely to be aligned, whereas inconsistent ones are potentially misaligned and may introduce bias. Building on this intuition, we propose REFINE (Robust sEmantic sample Filtering for partIally view-aligNed clustEring), a semantic filtering framework that adaptively selects reliable samples, thereby mitigating the influence of misalignment and enhancing clustering robustness.

Specifically, REFINE processes each view through dedicated Siamese encoders to generate query and key embeddings, with query features subsequently projected into a shared latent space via a cross-view decoder. A view-specific clustering head is assigned to each view. The training process commences with a warm-up phase where encoders and decoders are pretrained using intra-view and inter-view contrastive losses. Subsequently, we periodically re-initialize cluster prototypes. We perform $k$-means on key embeddings to obtain initial pseudo-labels and identify semantically consistent samples. Then, we initialize more robust prototypes based on the identified reliable samples while constructing an inter-view class-level alignment matrix. These prototypes are embedded as learnable parameters for soft assignment computation via Student's t-distribution. Then, cross-modal semantic consistency is enforced through symmetric KL divergence minimization guided by the alignment matrix. To prevent degenerate solutions, an entropy regularization term is incorporated to discourage cluster collapse. Finally, we conduct shared-space contrastive learning that leverages spectrally denoised feature similarities to further enhance representations, ultimately yielding robust clustering performance despite partial view alignment. Our contributions are summarized as follows:

(1) Inspired by sample selection paradigm in NLL, we propose a cross-view semantics-based filtering strategy to filter unreliable samples during cluster prototype initialization and cross-modal semantic consistency learning. This strategy mitigates noise interference by eliminating semantically inconsistent samples, boosting model robustness in partially view-aligned scenarios.

(2) We design a shared cross-view decoder architecture that projects heterogeneous view-specific features into a unified latent space. This design promotes representation consistency across views, enabling more effective cross-view contrastive learning.

(3) We conduct extensive experiments across five benchmark datasets under both fully and partially aligned settings. Our comprehensive evaluation demonstrates that REFINE achieves state-of-the-art performance, demonstrating its effectiveness across diverse scenarios.

## 2 RELATED WORK

### 2.1 MULTI-VIEW CLUSTERING (MVC)

**Complete MvC**: In the complete MvC setting, all samples are fully observed with perfectly aligned views across all modalities. Early studies mainly focus on conventional approaches such as non-negative matrix factorization (NMF) (Guan et al., 2012), graph-based methods (Chen et al., 2020a),

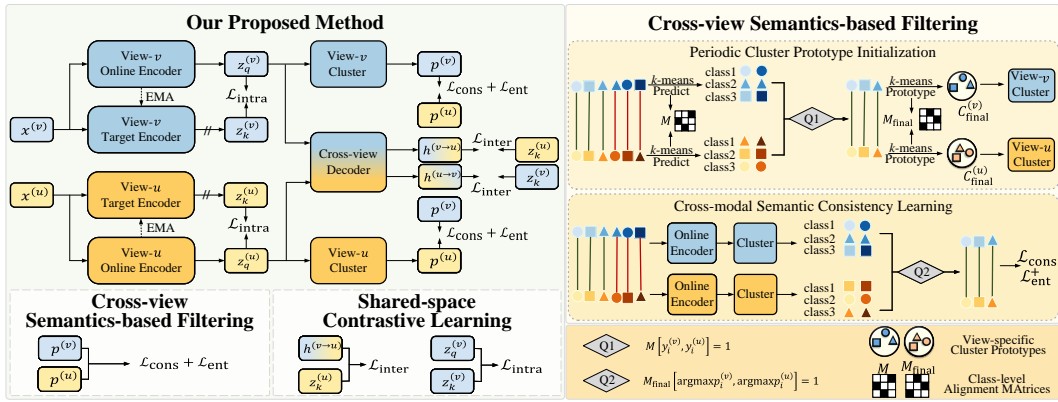

Figure 2: Overview of our REFINE. The left part illustrates the overall framework. Each view passes through the view-specific Siamese encoders to produce embeddings $z_q$ and $z_k$. The shared cross-view decoder then projects $z_q$ into a unified latent space. Meanwhile, a view-specific clustering module produces soft semantic predictions based on $z_q$. The overall objective includes: (1) consistency learning with entropy regularization on reliable samples with aligned pseudo-labels (Cross-view Semantics-based Filtering), and (2) intra-view and inter-view denoising contrastive losses (Shared-space Contrastive Learning). The right part illustrates details of the Cross-view Semantics-based Filtering module, which supports both prototype initialization and consistency learning by identifying semantically reliable sample pairs.

and subspace clustering (Cao et al., 2015; Li et al., 2019). However, these methods typically rely on shallow and linear embeddings, limiting their capability to capture complex nonlinear relationships in multi-view data. Recent research has shifted toward deep learning-based approaches (Trosten et al., 2023; Wang et al., 2025), where deep networks enable more powerful feature extraction and have consequently achieved remarkable clustering performance.

**Incomplete MvC (IMvC)**: In the IMvC setting, a subset of views may be missing for certain samples (Yu et al., 2025; Gao & Pu, 2025). Existing methods typically leverage the available complete views to predict or impute the missing data. For example, DIVIDE (Lu et al., 2024) enhances clustering under partial view-missing circumstances by using high-order random walks to globally identify reliable sample pairs. SMILE (Zeng et al., 2023) tackles IMvC by exploiting invariant semantic distributions across views, enabling robust imputation.

**Partially View-aligned Clustering (PVC)**: In this setting, multiple views corresponding to the same instance may exhibit misalignment. Most existing methods assume the aligned samples are known and use them to establish correspondences for realignment and clustering. For example, VI-TAL (He et al., 2024) disentangles shared and view-specific information via Gaussian modeling and variational contrastive learning, while Wang et al. (2024a) employs graph matching to infer correspondences from similarity graphs. However, this assumption of known aligned pairs often fails in practice. Recent advances have addressed this limitation by developing more flexible approaches that can accommodate unaligned samples during training. For example, CANDY (Guo et al., 2024) mitigates false negatives and positives via contextual similarity and a spectral module. ROLL (Sun et al., 2025) designs specific losses for noisy pseudo-labels and correspondences. However, most previous methods rely on robust loss functions to counter noise, which may risk underfitting reliable correspondences. In contrast, we adopt a sample-filtering strategy leveraging semantic information to discard unreliable samples and fully learn from trustworthy ones, improving clustering robustness.

## 2.2 NOISY LABEL LEARNING (NLL)

Recent years have witnessed significant research efforts devoted to developing robust methods capable of handling noisy labels (Cheng et al., 2021; Sun et al., 2022; Sheng et al., 2024). Existing approaches can be broadly classified into three main categories: (1) loss-based methods that enhance noise robustness through carefully designed loss functions (Li et al., 2024; Zhou et al., 2023); (2) sample-based methods that employ mechanisms to discriminate between clean and noisy sam-

ples (Xia et al., 2022; Nguyen et al., 2020); and (3) label-based methods that focus on correcting erroneous labels (Zhou et al., 2021; Li et al., 2023).

Among these, sample-based NLL methods often rely on a selection strategy that identifies a subset of samples with high-confidence predictions, which are then used to guide training while avoiding the influence of noisy labels (Li et al., 2023). For example, Jo-SRC (Yao et al., 2021) selects clean samples using a Jensen-Shannon (JS) divergence metric. DivideMix (Lu et al., 2024) distinguishes training samples using a dual-module Gaussian Mixture Model (GMM). Our work addresses a specific form of label noise where errors predominantly stem from incorrect sample correspondences across different views. Drawing inspiration from sample selection strategies in noisy label learning (NLL), we utilize semantic consistency between paired samples as our primary criterion - samples exhibiting inconsistent predicted cluster assignments are identified as potentially noisy pairs.

## 3 METHOD

**Preliminaries**: Given a multi-view clustering dataset $D = \{(x_i^{(1)}, \ldots, x_i^{(V)})\}_{i=1}^N$ with $N$ instances from $V$ views, conventional MvC methods assume complete alignment across views and aim to group these instances into $K$ clusters. In contrast to this standard fully aligned MvC setting, we address the more challenging scenario of partially misaligned views. The unknown sample alignment status introduces substantial complexity to the clustering process.

To address the above issues, we propose a novel contrastive MvC method called REFINE, which comprises two key components: Cross-view Semantics-based Filtering and Shared-space Contrastive Learning. The overall framework is shown in Figure 2, and the detailed algorithm (pseudo-code) is illustrated in the supplementary material. For simplicity, we illustrate with two views.

### 3.1 CROSS-VIEW SEMANTICS-BASED FILTERING

Drawing inspiration from sample selection in noisy label learning, we propose a Cross-view Semantics-based Filtering mechanism that simultaneously guides both prototype initialization and cross-view consistency learning, thereby minimizing the effects of misaligned or unreliable samples.

**Warmup**: We begin by warming up the network through contrastive learning to obtain discriminative feature representations. This initialization facilitates robust sample filtering in subsequent cluster prototype initialization and cross-view consistency learning phases. Both intra- and inter-view contrastive objectives are applied to all samples in this stage.

Specifically, each view employs the view-specific Siamese encoders, consisting of an online encoder $f_q$ and a momentum-updated key encoder $f_k$. Given an instance $x_i^{(v)}$ from view $v$, we obtain correspondence embeddings:

$$z_{q,i}^{(v)} = f_q^{(v)}(x_i^{(v)}), \ \ z_{k,i}^{(v)} = f_k^{(v)}(x_i^{(v)}). \tag{1}$$

Then we adopt a shared cross-view decoder $g$, which maps the view-specific embedding $z_{q,i}^{(v)}$ into a shared latent space across all views as follows:

$$h_i^{(v \to u)} = g(z_{q,i}^{(v)}). \tag{2}$$

$z_{q,i}^{(v)}$, $z_{k,i}^{(v)}$, and $h_i^{(v \to u)}$ have same dimensionality $d$. Similarly, we can obtain $z_{q,i}^{(u)}$, $z_{k,i}^{(u)}$, and $h_i^{(u \to v)}$.

We calculate intra-view and inter-view contrastive losses based on NT-Xent loss (Chen et al., 2020b):

$$\mathcal{L}_{\text{intra}} = \frac{1}{2N_b}(\mathcal{H}(\mathbf{I}, \rho(z_q^{(v)}, z_k^{(v)})) + \mathcal{H}(\mathbf{I}, \rho(z_q^{(u)}, z_k^{(u)}))),$$

$$\mathcal{L}_{\text{inter}} = \frac{1}{2N_b}(\mathcal{H}(\mathbf{I}, \rho(h^{(v \to u)}, z_k^{(u)})) + \mathcal{H}(\mathbf{I}, \rho(h^{(u \to v)}, z_k^{(v)}))). \tag{3}$$

$N_b$ is the mini-batch size. $\mathcal{H}(\cdot, \cdot)$ denotes the cross entropy. $\mathbf{I} \in R^{N_b \times N_b}$ is an identity matrix indicating hard alignment (with 1 on diagonal and 0 elsewhere). $\rho(a, b)$ is the pairwise similarity with the row-wise normalization operator:

$$[\rho(a, b)]_{ij} = -\log \frac{\exp\left(\frac{\text{sim}(a_i, b_j)}{\tau}\right)}{\sum_{l=1}^{N_b} \exp\left(\frac{\text{sim}(a_i, b_l)}{\tau}\right)}. \tag{4}$$

$\text{sim}(\cdot, \cdot)$ represents the cosine similarity between embeddings, and $\tau$ is a temperature hyperparameter (which is set to 0.5 by default). Finally, the overall loss is defined as:

$$\mathcal{L}_{\text{warm}} = \mathcal{L}_{\text{intra}} + \mathcal{L}_{\text{inter}}. \tag{5}$$

**Periodic Cluster Prototype Initialization**: Following the warm-up phase, the model acquires moderately discriminative feature representations. We subsequently implement periodic reliable sample filtering to establish high-quality cluster prototypes. This process begins by independently clustering the embeddings from views $v$ and $u$ to obtain their respective cluster centroids:

$$C^{(v)} = k\text{-means}(Z^{(v)}), C^{(u)} = k\text{-means}(Z^{(u)}), \tag{6}$$

where $Z^{(v)} = \{z_{k,i}^{(v)}\}_{i=1}^{N}$, $Z^{(u)} = \{z_{k,i}^{(u)}\}_{i=1}^{N}$. $C^{(v)}, C^{(u)} \in \mathbb{R}^{K \times d}$ represent the $K$ cluster prototypes in the $d$-dimensional embedding space of views $v$ and $u$.

Secondly, we employ the Hungarian algorithm (Kuhn, 1955) to align clusters across views, producing a class-level alignment matrix $M \in \{0, 1\}^{K \times K}$. We select samples with consistent predictions:

$$I_{\text{align}} = \{i \in \{1, \ldots, N\} \mid M[y_i^{(v)}, y_i^{(u)}] = 1\}. \tag{7}$$

$y_i^{(v)}$ and $y_i^{(u)}$ denote the pseudo-labels of instance $i$ obtained from $C^{(v)}$ and $C^{(u)}$, respectively. An entry $M[a, b] = 1$ indicates that cluster $a$ in view $v$ is aligned with cluster $b$ in view $u$; otherwise, $M[a, b] = 0$.

Finally, we leverage selected consistent samples to refine cluster prototypes, enhancing reliability:

$$C_{\text{final}}^{(v)} = k\text{-means}(\{z_{k,j}^{(v)} \mid j \in I_{\text{align}}\}), \ \ C_{\text{final}}^{(u)} = k\text{-means}(\{z_{k,j}^{(u)} \mid j \in I_{\text{align}}\}). \tag{8}$$

$C_{\text{final}}^{(v)}$ and $C_{\text{final}}^{(u)}$ are used to initialize the view-specific clustering heads. The final class-level alignment matrix $M_{\text{final}}$ is obtained via the Hungarian algorithm on $N$ instances.

**Cross-modal Semantic Consistency Learning**: We utilize the cluster prototypes to assign pseudo-labels and guide consistency learning. Specifically, for $x_i^{(v)}$, we compute the soft pseudo-label $p_i^{(v)}$ using Student's t-distribution (Xie et al., 2016; Ghasedi Dizaji et al., 2017):

$$p_{ik}^{(v)} = \frac{(1 + \frac{|z_{q,i}^{(v)} - c_k^{(v)}|^2}{\alpha})^{-\frac{\alpha+1}{2}}}{\sum_{c_{k'}^{(v)} \in C^{(v)}} (1 + \frac{|z_{q,i}^{(v)} - c_{k'}^{(v)}|^2}{\alpha})^{-\frac{\alpha+1}{2}}}. \tag{9}$$

$p_{ik}^{(v)}$ denotes the probability of assigning instance $i$ in view $v$ to cluster $k$. $c_k^{(v)}$ is the prototype of cluster $k$ in view $v$. $\alpha$ is a hyperparameter (set to 1 by default). Similarly, we can obtain $p_i^{(u)}$. Then, we identify a set of trusted samples whose pseudo-labels from different views are aligned:

$$I_{\text{trusted}} = \{i \in B \mid M_{\text{final}}[\arg\max p_i^{(v)}, \arg\max p_i^{(u)}] = 1\}, \tag{10}$$

where $B$ denotes the set of indices in the current mini-batch.

For these aligned samples, we enforce consistency by minimizing the symmetric KL divergence (Kullback & Leibler, 1951) between the view-specific soft pseudo-labels:

$$\mathcal{L}_{\text{cons}} = \frac{1}{2|I_{\text{trusted}}|} \sum_{j}^{I_{\text{trusted}}} (\text{KL}(p_j^{(u)}|p_j^{(v \to u)}) + \text{KL}(p_j^{(v \to u)}|p_j^{(u)})), \tag{11}$$

where $p_j^{(v \to u)} = p_j^{(v)} M_{\text{final}}$ denotes the cross-view aligned label distribution. This loss enforces cross-view semantic consistency by aligning the predicted label distributions of trusted instances.

To prevent cluster collapse, we further incorporate an entropy regularization term (Caron et al., 2020; Van Gansbeke et al., 2020) on soft assignment distributions. Specifically, we compute the marginal cluster distribution as:

$$\pi(k) = \frac{1}{|I_{\text{trusted}}|} \sum_{j}^{I_{\text{trusted}}} p_{jk}^{(v \to u)} + p_{jk}^{(u)}. \tag{12}$$

The overall entropy regularization loss is then calculated as:

$$\mathcal{L}_{\text{ent}} = -\sum_{k=1}^{K} \pi(k) \log(\pi(k) + \epsilon). \tag{13}$$

Here, we add a small constant $\epsilon = 10^{-8}$ for numerical stability to prevent logarithm-of-zero. This formulation promotes a uniform marginal distribution across clusters through entropy maximization.

**Discussion**: Inspired by the sample selection paradigm in NLL, we incorporate the Cross-view Semantics-based Filtering into both cluster prototype initialization and consistency learning. Through joint noise filtering across these two stages, we reduce the influence of misaligned samples, yielding more reliable prototype construction and more robust cross-view supervision.

## 3.2 SHARED-SPACE CONTRASTIVE LEARNING

Following (Guo et al., 2024; Lu et al., 2024), we construct soft contrastive losses within and across views based on feature similarity, while employing spectral denoising to enhance correspondence quality. Anchoring on view $v$ (*i.e.*, updating view $v$'s parameters only), we compute intra-view and inter-view affinities as:

$$\mathbf{A}_{ij}^{(u \to u)} = \begin{cases} 1, & i = j \\ \exp(-\frac{\|h_i^{(v \to u)} - z_{k,j}^{(u)}\|^2}{\sigma}), & i \neq j \end{cases}, \tag{14}$$

$$\mathbf{A}_{ij}^{(v \to u)} = \exp(-\frac{\|h_i^{(v \to u)} - z_{k,j}^{(u)}\|^2}{\sigma}), \quad i, j \in B.$$

$\sigma$ is a hyperparameter that is set to 0.07 by default. We then construct high-order affinity graphs by combining intra-view and inter-view affinities as follows:

$$\mathbf{G}_{\text{intra}}^{(v)} = \mathbf{A}^{(u \to u)} \mathbf{A}^{(u \to u)^\top}, \quad \mathbf{G}_{\text{inter}}^{(v)} = \mathbf{A}^{(v \to u)} \mathbf{A}^{(u \to u)^\top}. \tag{15}$$

Subsequently, we perform singular value decomposition (SVD) (Eckart & Young, 1936; Rajwade et al., 2012) to selectively discard information corresponding to small singular values, thereby filtering out noise. For $\mathbf{G}^{(v)} = \mathbf{U}\boldsymbol{\Sigma}\mathbf{V}^\top$, where $\boldsymbol{\Sigma}$ is a diagonal matrix containing the singular values and $\mathbf{U}, \mathbf{V}$ are the left and right singular matrices respectively, we retain only the top singular values above a threshold $\eta$. Specifically, the noise-resistant high-order affinity graph is reconstructed by:

$$\widetilde{\mathbf{G}}^{(v)} = \mathbf{U}\widetilde{\boldsymbol{\Sigma}}\mathbf{V}^\top, \widetilde{\boldsymbol{\Sigma}} = \text{diag}(\lambda_1, \ldots, \lambda_L), \lambda_1 > \cdots > \lambda_L \geq \eta. \tag{16}$$

We then construct the soft pseudo targets derived from intra- and inter-view affinity graphs:

$$\mathbf{C}_{\text{intra}}^{(v)} = \lambda \mathbf{I} + \widetilde{\mathbf{G}}_{\text{intra}}^{(v)}, \quad \mathbf{C}_{\text{inter}}^{(v)} = \lambda \mathbf{I} + \widetilde{\mathbf{G}}_{\text{inter}}^{(v)}, \tag{17}$$

where $\lambda$ is fixed as 0.2 by default. Similarly, we can compute $\mathbf{C}_{\text{intra}}^{(u)}$ and $\mathbf{C}_{\text{inter}}^{(u)}$ anchoring on view $u$.

Lastly, intra- and inter-view soft contrastive losses are:

$$\mathcal{L}_{\text{intra}}^{\text{soft}} = \frac{1}{2N_b}(\mathcal{H}(\mathbf{C}_{\text{intra}}^{(v)}, \rho(z_q^{(v)}, z_k^{(v)})) + \mathcal{H}(\mathbf{C}_{\text{intra}}^{(u)}, \rho(z_q^{(u)}, z_k^{(u)}))),$$

$$\mathcal{L}_{\text{inter}}^{\text{soft}} = \frac{1}{2N_b}(\mathcal{H}(\mathbf{C}_{\text{inter}}^{(v)}, \rho(h^{(v \to u)}, z_k^{(u)})) + \mathcal{H}(\mathbf{C}_{\text{inter}}^{(u)}, \rho(h^{(u \to v)}, z_k^{(v)}))). \tag{18}$$

All together, our final loss is:

$$\mathcal{L}_{\text{total}} = \mathcal{L}_{\text{intra}}^{\text{soft}} + \mathcal{L}_{\text{inter}}^{\text{soft}} + \gamma_1 \mathcal{L}_{\text{cons}} + \gamma_2 \mathcal{L}_{\text{ent}}, \tag{19}$$

where $\gamma_1$ and $\gamma_2$ control the consistency and entropy terms.

**Discussion**: Unlike previous works (Guo et al., 2024; Lu et al., 2024) that conduct contrastive learning in distinct view-specific spaces, our approach utilizes a shared cross-view decoder to project all views into a unified feature space. This design yields two key advantages: (1) narrowing the modality gap between views enhances contrastive learning effectiveness; (2) improved feature consistency across views enables more accurate semantics-based filtering. These benefits mutually reinforce each other during optimization.

**Scalability to More Views:** When $V > 2$, class-level alignment matrices are constructed by selecting a reference view and computing the alignment between this reference and each of the other views. The alignment matrices between the non-reference views can then be obtained indirectly by composing them via matrix multiplication. This design allows our method to be naturally extended to scenarios beyond two views, making it applicable to general multi-view clustering tasks.

## 4 EXPERIMENTS

### 4.1 DATASETS AND BASELINES

We validate REFINE on five datasets, consisting of Scene15 (Fei-Fei & Perona, 2005) (4,485 samples with 15 classes), Caltech101 (Li et al., 2015) (8,677 samples with 101 classes), LandUse-21 (Li et al., 2015) (2,100 samples with 21 classes), Reuters (Amini et al., 2009) (18,758 samples with 6 classes), and NUS-WIDE (Hu et al., 2019) (9,000 samples with 10 classes). Following Guo et al. (2024), we simulate partial view misalignment by selecting one view as the anchor and randomly shuffling the samples in the other views according to the predefined false positive (FP) ratio.

The following methods are used as baselines to illustrate the effectiveness of our proposed RE-FINE: Complete MvC methods (*i.e.*, DCCAE (Wang et al., 2015), BMvC (Zhang et al., 2018)), Incomplete Multi-view Clustering (IMvC) methods (*i.e.*, GCFAgg (Yan et al., 2023), DIVIDE (Lu et al., 2024)), and Partially View-aligned Clustering (PVC) methods (*i.e.*, MvCLN (Yang et al., 2021), PVC (Huang et al., 2020), SURE (Yang et al., 2022), CGCN (Wang et al., 2024b), and CANDY (Guo et al., 2024)).

### 4.2 IMPLEMENTATION DETAILS

As shown in Figure 2, REFINE consists of three learnable modules: view-specific Siamese encoder, shared cross-view decoder, and view-specific cluster layer. For the encoder and decoder, we use fully-connected networks (FCNs), where each layer is followed by a BatchNorm layer, a ReLU layer, and Dropout. The final BatchNorm layer of the encoder does not include learnable affine parameters. The cross-view decoder is a two-layer MLP, and the cluster layer contains learnable centroids. Further details are provided in the supplementary material.

To assess the effectiveness of our method, we utilize "ACC", "NMI", and "ARI" as evaluation metrics. Reported results are the averages of five repeated experiments conducted under identical settings. Our method is implemented using PyTorch 2.5.1 on NVIDIA RTX 4090 GPUs. Adam is utilized as the optimizer. Key hyperparameter settings are provided in Table 1. The remaining settings (*i.e.*, those explicitly specified in Method) follow CANDY (Guo et al., 2024).

Table 1: Experiment Hyperparameters.

| Dataset | Lr | Epoch (Warmup) | Batch | $e_f$ | $\eta$ | $\gamma_1$ | $\gamma_2$ |
|---|---|---|---|---|---|---|---|
| Scene15 | 0.004 | 200 (20) | 1024 | 20 | 0.4 | 1.0 | 1.0 |
| Caltech-101 | 0.0004 | 100 (20) | 1024 | 20 | 0.4 | 10.0 | 1.0 |
| LandUse-21 | 0.004 | 100 (20) | 1024 | 20 | 0.4 | 10.0 | 1.0 |
| Reuters | 0.0004 | 200 (20) | 1024 | 20 | 0.4 | 1.0 | 1.0 |
| NUS-WIDE | 0.0004 | 100 (20) | 1024 | 20 | 0.4 | 1.0 | 1.0 |

### 4.3 EXPERIMENTAL RESULTS

Table 2 presents the experimental comparisons across five benchmark datasets. Performance results of competing methods are primarily sourced from Guo et al. (2024). All methods are evaluated using identical input features and preprocessing for each dataset. From this table, we have the following observations. First, when FP ratio is 0% (which means all sample views are perfectly aligned), REFINE outperforms baseline methods in certain scenarios. These performances validate that our sample selection strategy effectively identifies more representative aligned samples, leading to improved clustering quality. Second, when handling misaligned views (*i.e.*, FP ratio 20%, 50%, or 80%), most baseline methods suffer significant performance degradation. In contrast, RE-FINE exhibits relatively robustness. Notably, in terms of average performance, we still outperform all baseline methods across all noise levels. This confirms that our sample selection mechanism successfully filters misaligned samples while maintaining strong noise resistance.

### 4.4 ABLATION STUDIES

Table 2: Comparison on five benchmark benchmark datasets. The best and second-best results are shown in **bold** and underline, respectively.

| FP Ratio | Methods | Scene15 | | | Caltech-101 | | | LandUse21 | | | Reuters | | | NUS-WIDE | | | Average | | |
|---|---|---|---|---|---|---|---|---|---|---|---|---|---|---|---|---|---|---|---|
| | | ACC | NMI | ARI | ACC | NMI | ARI | ACC | NMI | ARI | ACC | NMI | ARI | ACC | NMI | ARI | ACC | NMI | ARI |
| 0% | DCCAE (ICML'15) | 34.6 | 39.0 | 19.7 | 45.8 | 68.6 | 37.7 | 15.6 | 24.4 | 4.4 | 42.0 | 20.3 | 8.5 | 47.5 | 17.1 | 37.6 | 37.1 | 33.9 | 21.6 |
| | BMvC (TPAMI'18) | 40.5 | 41.2 | 24.1 | 50.1 | 72.4 | 33.9 | 25.3 | 28.6 | 11.4 | 42.4 | 21.9 | 15.1 | 36.0 | 21.0 | 16.5 | 38.9 | 37.0 | 20.2 |
| | PVC (NeurIPS'20) | 38.0 | 39.8 | 21.1 | 20.5 | 51.4 | 15.7 | 16.8 | 25.2 | 5.6 | 44.1 | 27.1 | 27.1 | 19.3 | 7.7 | 3.8 | 27.7 | 30.2 | 14.7 |
| | MvCLN (CVPR'21) | 37.9 | 42.3 | 25.6 | 39.6 | 65.3 | 32.8 | 26.1 | 30.7 | 12.5 | 38.8 | 42.1 | 25.2 | 54.1 | 38.3 | 35.7 | 39.3 | 43.7 | 26.4 |
| | SURE (TPAMI'23) | 41.0 | 43.2 | 25.0 | 43.8 | 70.1 | 29.5 | 25.1 | 28.3 | 10.9 | 49.1 | 29.9 | 23.6 | 57.4 | 44.8 | 38.3 | 43.3 | 43.3 | 25.5 |
| | GCFAgg (CVPR'23) | 42.2 | 42.5 | 24.4 | 56.6 | 80.7 | 37.9 | 27.5 | 31.3 | 14.0 | 34.4 | 23.8 | 10.5 | 41.1 | 32.1 | 18.6 | 40.4 | 42.1 | 21.1 |
| | CGCN (TCSVT'24) | 42.9 | 43.4 | 25.0 | 49.1 | 75.2 | 33.8 | 28.8 | 36.0 | 15.0 | 45.8 | 27.0 | 22.3 | 61.2 | 48.1 | 41.2 | 45.6 | 45.9 | 27.5 |
| | DIVIDE (AAAI'24) | **49.1** | **48.7** | **31.6** | 62.2 | 83.0 | 50.5 | 32.3 | **39.7** | **18.1** | 59.3 | 39.5 | **29.0** | 45.1 | 30.9 | 19.4 | 49.6 | 48.4 | 29.7 |
| | CANDY (NeurIPS'24) | 42.0 | 41.6 | 24.7 | 67.3 | 83.8 | 60.0 | 30.5 | 35.4 | 15.8 | 57.7 | 30.8 | **37.1** | 62.1 | 49.0 | 37.0 | 51.9 | 48.3 | 35.0 |
| | RMCNC (TKDE'24) | 43.7 | 43.3 | 26.3 | 34.6 | 42.0 | 51.0 | - | - | - | - | - | - | 67.6 | 53.6 | 49.0 | - | - | - |
| | DDMVC (PR'25) | 42.8 | 43.5 | - | - | - | - | - | - | - | 55.0 | 28.6 | - | - | - | - | - | - | - |
| | MCMVC (TPAMI'25) | 42.8 | 46.6 | 26.7 | - | - | - | - | - | - | - | - | - | - | - | - | - | - | - |
| | REFINE | 44.4 | 46.0 | 26.1 | **68.3** | **84.8** | **72.9** | **32.5** | 39.0 | **18.1** | 52.1 | 34.1 | 28.5 | **67.0** | **54.3** | **49.5** | **52.9** | **51.6** | **39.0** |
| 20% | DCCAE (ICML'15) | 32.9 | 17.1 | **29.6** | 36.9 | 39.2 | 60.1 | 15.0 | 3.8 | 17.4 | 41.6 | 13.1 | 19.3 | 41.6 | 11.6 | 26.9 | 33.6 | 17.0 | 30.7 |
| | BMvC (TPAMI'18) | 20.0 | 10.2 | 4.7 | 42.7 | 58.2 | 24.6 | 16.1 | 13.0 | 4.3 | 36.4 | 11.9 | 8.1 | 27.7 | 10.7 | 7.7 | 28.6 | 20.8 | 9.9 |
| | PVC (NeurIPS'20) | 31.2 | 25.5 | 13.6 | 8.3 | 30.2 | 3.8 | 22.8 | 28.0 | 8.4 | 32.4 | 15.4 | 15.3 | 34.3 | 22.2 | 13.6 | 25.8 | 24.3 | 10.9 |
| | MvCLN (CVPR'21) | 39.3 | 36.7 | 21.7 | 43.3 | 64.0 | 52.8 | 24.4 | 26.1 | 10.8 | 37.9 | 35.9 | 20.3 | 42.5 | 29.3 | 21.3 | 37.5 | 38.4 | 25.4 |
| | SURE (TPAMI'23) | 40.0 | 37.3 | 21.5 | 26.9 | 49.9 | 18.0 | 25.2 | 27.4 | 11.6 | 40.7 | 20.9 | 15.8 | 57.0 | 45.0 | 38.6 | 38.0 | 36.1 | 21.1 |
| | GCFAgg (CVPR'23) | 40.9 | 38.6 | 22.7 | 50.1 | 70.6 | 30.1 | 25.7 | 27.8 | 11.9 | 35.2 | 19.0 | 10.8 | 38.6 | 23.3 | 15.6 | 38.1 | 35.9 | 18.2 |
| | CGCN (TCSVT'24) | 40.7 | 38.0 | 22.1 | 40.8 | 64.9 | 27.2 | 27.0 | 31.4 | 13.3 | 43.5 | 23.0 | 19.4 | 58.0 | 41.7 | 35.9 | 42.0 | 39.8 | 23.6 |
| | DIVIDE (AAAI'24) | **42.4** | 39.9 | 24.5 | 48.3 | 69.1 | 38.0 | 30.9 | 35.1 | 16.2 | 55.3 | **36.9** | 31.0 | 44.9 | 28.3 | 18.2 | 44.4 | 41.9 | 25.6 |
| | CANDY (NeurIPS'24) | 40.4 | 40.3 | 23.7 | 65.9 | 82.3 | 60.1 | 30.5 | 35.3 | 15.7 | 54.2 | 27.9 | 33.8 | 60.3 | 47.1 | 36.9 | 50.3 | 46.6 | 34.0 |
| | REFINE | 41.7 | **43.1** | 24.0 | **66.7** | **83.8** | **72.3** | **31.9** | **38.1** | **17.5** | **55.9** | 32.5 | **33.9** | **66.9** | **52.3** | **48.2** | **52.6** | **50.0** | **39.2** |
| 50% | DCCAE (ICML'15) | 26.8 | 10.2 | 19.8 | 27.0 | 26.8 | 49.8 | 13.3 | 2.8 | 13.2 | 37.7 | 9.2 | 12.5 | 32.3 | 7.1 | 13.5 | 27.4 | 11.2 | 21.8 |
| | BMvC (TPAMI'18) | 13.6 | 3.9 | 1.4 | 26.5 | 34.2 | 8.9 | 13.5 | 7.5 | 1.9 | 26.3 | 2.3 | 2.3 | 18.4 | 3.1 | 1.9 | 19.7 | 10.4 | 3.3 |
| | PVC (NeurIPS'20) | 20.3 | 13.2 | 13.6 | 7.4 | 21.8 | 5.0 | 20.6 | 28.5 | 8.7 | 42.9 | 23.5 | 23.4 | 24.1 | 10.1 | 9.9 | 23.1 | 18.8 | 12.1 |
| | MvCLN (CVPR'21) | 41.3 | 19.7 | 15.1 | 21.4 | 39.1 | 11.7 | 21.4 | 21.8 | 7.8 | 34.8 | **35.5** | 19.7 | 31.7 | 16.6 | 10.7 | 30.1 | 26.5 | 13.0 |
| | SURE (TPAMI'23) | 37.1 | 35.7 | 20.3 | 19.9 | 41.7 | 13.2 | 23.1 | 22.8 | 8.9 | 38.0 | 18.5 | 14.3 | 35.0 | 17.4 | 12.0 | 30.6 | 27.2 | 13.7 |
| | GCFAgg (CVPR'23) | 34.1 | 32.9 | 17.3 | 42.2 | 63.0 | 24.8 | 25.2 | 24.9 | 10.9 | 28.5 | 8.9 | 4.5 | 26.7 | 10.5 | 6.4 | 31.3 | 28.0 | 12.8 |
| | CGCN (TCSVT'24) | 32.5 | 29.5 | 15.7 | 33.4 | 59.3 | 21.6 | 26.3 | 28.1 | 11.9 | 40.5 | 16.1 | 14.1 | 50.1 | 33.8 | 27.4 | 36.5 | 33.4 | 18.1 |
| | DIVIDE (AAAI'24) | 37.4 | 34.0 | 20.3 | 39.1 | 58.7 | 32.5 | 28.1 | 30.4 | 13.5 | 41.2 | 19.4 | 14.8 | 44.0 | 23.9 | 16.6 | 38.0 | 33.3 | 19.5 |
| | CANDY (NeurIPS'24) | 41.3 | **39.4** | **24.0** | 60.7 | 79.0 | 56.6 | 29.9 | 33.1 | 15.2 | 47.4 | 21.7 | 27.3 | 58.1 | 43.2 | 34.5 | 47.5 | 43.3 | 31.5 |
| | REFINE | 40.8 | 41.4 | 23.4 | **63.4** | **82.1** | **69.0** | **31.5** | **37.3** | **16.8** | **55.0** | 30.8 | **28.7** | **64.1** | **48.3** | **44.2** | **51.0** | **48.0** | **36.4** |
| 80% | DCCAE (ICML'15) | 20.9 | 6.7 | 14.4 | 18.4 | 15.8 | 41.8 | 14.5 | 3.2 | 13.4 | 35.3 | 7.6 | 10.0 | 36.2 | 14.9 | 21.9 | 25.1 | 9.6 | 20.3 |
| | BMvC (TPAMI'18) | 10.5 | 1.5 | 0.3 | 11.9 | 18.3 | 1.5 | 10.1 | 4.2 | 0.4 | 21.3 | 0.5 | 0.1 | 13.1 | 0.6 | 0.2 | 13.4 | 5.0 | 0.5 |
| | PVC (NeurIPS'20) | 20.3 | 10.2 | 4.6 | 7.5 | 20.8 | 4.2 | 22.5 | 29.3 | 9.3 | 35.7 | 13.2 | 13.1 | 19.3 | 7.7 | 3.8 | 21.1 | 16.2 | 7.0 |
| | MvCLN (CVPR'21) | 35.7 | 16.2 | 13.9 | 13.9 | 34.2 | 10.9 | 17.0 | 15.7 | 4.4 | 24.3 | **28.1** | 12.4 | 24.3 | 10.0 | 5.7 | 23.0 | 20.8 | 9.5 |
| | SURE (TPAMI'23) | 27.4 | 30.7 | 14.2 | 16.2 | 38.3 | 9.0 | 18.0 | 17.6 | 5.5 | 34.6 | 15.5 | 13.0 | 23.7 | 9.4 | 5.4 | 24.0 | 22.3 | 9.4 |
| | GCFAgg (CVPR'23) | 26.5 | 24.8 | 11.4 | 26.7 | 45.5 | 12.6 | 22.4 | 23.0 | 8.7 | 25.6 | 4.6 | 2.7 | 17.0 | 3.0 | 1.5 | 23.6 | 20.2 | 7.4 |
| | CGCN (TCSVT'24) | 28.7 | 24.0 | 12.5 | 21.3 | 46.6 | 13.2 | 25.2 | 27.7 | 11.4 | 29.0 | 7.9 | 6.5 | 50.1 | 34.6 | 28.0 | 30.9 | 28.2 | 14.3 |
| | DIVIDE (AAAI'24) | 34.4 | 30.4 | 18.3 | 27.8 | 50.8 | 21.1 | 27.1 | 28.1 | 12.8 | **41.1** | 24.7 | **19.5** | 45.8 | 28.3 | 19.1 | 35.2 | 32.5 | 18.2 |
| | CANDY (NeurIPS'24) | 38.8 | 36.6 | 20.7 | 52.6 | **76.8** | 52.9 | 28.1 | 31.3 | 13.5 | 37.0 | 12.4 | 15.6 | 55.6 | 39.1 | 32.6 | 42.4 | 39.2 | 27.1 |
| | REFINE | **40.8** | **39.7** | **23.2** | **54.6** | 76.5 | **64.6** | **30.1** | **34.5** | **15.5** | 39.1 | 15.5 | 13.6 | **57.2** | **41.8** | **36.9** | **44.4** | **41.6** | **30.8** |

**Study of each component**: We validate the effectiveness of each component in our framework on Caltech-101 and NUS-WIDE under 20% and 50% FP ratios. The experimental results are shown in Table 3. "CSF" indicates the cross-view semantics-based filtering, which comprises three key components: (1) "Warmup", which refers to training using only hard intra-view and inter-view contrastive losses; (2) "PCPI", which stands for periodic cluster prototype initialization; (3) "CSCL", which denotes

Table 3: Study of each component on Caltech-101 and NUS-WIDE datasets under 20% and 50% FP ratios.

| FP Ratio | CSF | | | SCL | Caltech-101 | | | NUS-WIDE | | |
|---|---|---|---|---|---|---|---|---|---|---|
| | Warmup | PCPI | CSCL | | ACC | NMI | ARI | ACC | NMI | ARI |
| 20% | ✓ | | | | 47.7 | 69.0 | 30.3 | 55.2 | 34.8 | 30.2 |
| | ✓ | ✓ | ✓ | | 46.3 | 68.1 | 49.1 | 34.4 | 32.0 | 21.9 |
| | ✓ | | | ✓ | 67.1 | 81.6 | 64.7 | 62.5 | 50.1 | 39.6 |
| | ✓ | ✓ | ✓ | ✓ | 66.7 | 83.8 | 72.3 | 66.9 | 52.3 | 48.2 |
| 50% | ✓ | | | | 39.6 | 57.2 | 24.8 | 41.8 | 18.5 | 14.8 |
| | ✓ | ✓ | ✓ | | 45.4 | 67.1 | 47.6 | 26.6 | 17.8 | 11.6 |
| | ✓ | | | ✓ | 62.8 | 80.9 | 59.2 | 61.3 | 46.4 | 38.4 |
| | ✓ | ✓ | ✓ | ✓ | 63.4 | 82.1 | 69.0 | 64.1 | 48.3 | 44.2 |

cross-modal semantic consistency learning. Note that CSCL operates in conjunction with PCPI, as it requires up-to-date cluster prototypes for reliable pseudo-label supervision. Additionally, "SCL" corresponds to the shared-space contrastive learning. From this study, we observe that the combination "Warmup+PCPI+CSCL" exhibits a significant performance degradation, performing worse than using "Warmup" alone. We attribute this to the training instability caused by abruptly introducing consistency-based losses after the warmup stage, where features have already been optimized purely through contrastive losses. In contrast, the full framework "Warmup+PCPI+CSCL+SCL" outperforms "Warmup+SCL". This indicates that the feature space becomes more robust when consistency regularization is properly integrated. Moreover, it affirms that SCL provides the necessary stabilization for CSCL's consistency constraints.

These results demonstrate that each component contributes distinct and complementary benefits to our REFINE, while their systematic integration proves essential for achieving optimal performance.

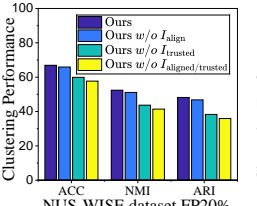 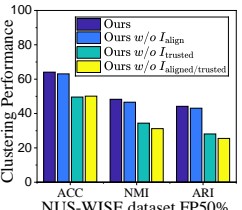 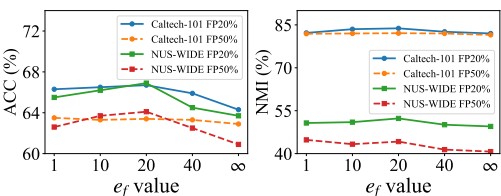

(a) Clustering performances of our filtering mechanism on NUS-WIDE under 20% and 50% FP ratios.

(b) Study of $e_f$ across the Caltech-101 and NUS-WIDE datasets under 20% and 50% FP ratios.

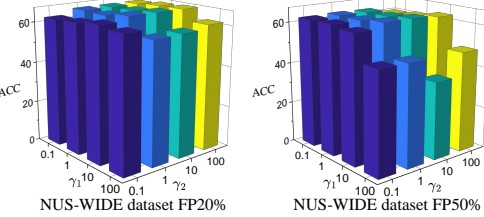 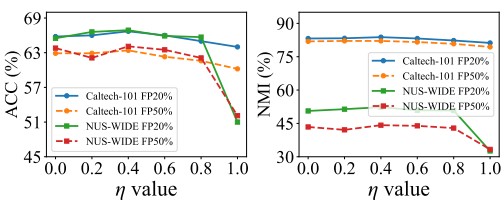

(c) Parameter sensitivity of $\gamma_1$ and $\gamma_2$ on the NUS-WIDE dataset under 20% and 50% FP ratios.

(d) Parameter sensitivity of $\eta$ on the Caltech-101 and NUS-WIDE datasets under 20% and 50% FP ratios.

Figure 3: Experimental results of ablation studies.

**Analysis of filtering mechanism**: We investigate the clustering accuracy of our proposed Cross-view Semantics-based Filtering on NUS-WIDE under 20% and 50% FP ratios in Figure 3 (a). "Ours" refers to the complete implementation of our method. "Ours $w/o\ I_{\text{align}}$" denotes the variant where sample filtering is disabled during prototype initialization. "Ours $w/o\ I_{\text{trusted}}$" represents the case where filtering is omitted in semantic consistency learning. "Ours $w/o\ I_{\text{align/trusted}}$" indicates the setting where both filtering mechanisms are excluded. The experimental results demonstrate that both filtering strategy contributes positively to clustering performance. Notably, the $I_{\text{trusted}}$-based filtering proves particularly effective by significantly reducing the impact of misaligned samples. These findings validate that our proposed filtering mechanism successfully adapts NLL sample selection principles to the PVC scenario.

**Analysis of filtering precision:** We evaluate the effectiveness of Cross-view Semantics-based Filtering strategy on NUS-WIDE under 20% and 50% FP ratios in Figure 4. Specifically, $I_{\text{aligned}}$ is the initial filtered set (Eq. (7)), directly affected by misaligned samples. $I_{\text{trusted}}$ is derived from the prediction probabilities of the prototype classifier (Eq. (10))—whose prototypes are initialized via $k$-means on $I_{\text{aligned}}$—thereby further reducing noise. Results show that REFINE effectively removes a

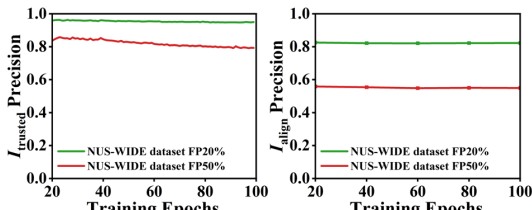

Figure 4: Filtering precision on NUS-WIDE.

large portion of misaligned samples while maintaining high precision for $I_{\text{trusted}}$. In more challenging 50% FP case, $I_{\text{aligned}}$ precision slightly drops due to high proportion of unaligned sample, but $I_{\text{trusted}}$ remains relatively robust. Overall, these findings confirm that our filtering mechanism successfully transfers NLL sample selection to the PVC setting, enabling stable and accurate multi-view clustering even under severe misalignment.

**Robustness analysis of NLL-inspired filtering:** Our method exploits the intuition that models tend to fit clean patterns before memorizing noisy ones, so semantic consistency across views can identify reliable samples. We validate this intuition on NUS-WIDE under different FP ratios, tracking clustering accuracy during training in Figure 5. We compare three variants: (1) "REFINE", which applies our full method; (2) "REFINE $w/o$ Filtering", which uses our method but without filtering (*i.e.*, all samples are used in both periodic cluster prototype initialization and cross-modal semantic consistency learning); and (3)"Base", which trains throughout using only warmup losses. The results show that: Firstly, as FP ratio increases, "Base" tends to overfit misaligned samples, degrading performance. Secondly, "REFINE $w/o$ Filtering" benefits from semantic learning but remains

unstable under high FP ratio. Lastly, "REFINE" adaptively filters unreliable samples, mitigating misalignment effects and achieving more robust clustering under partially aligned views.

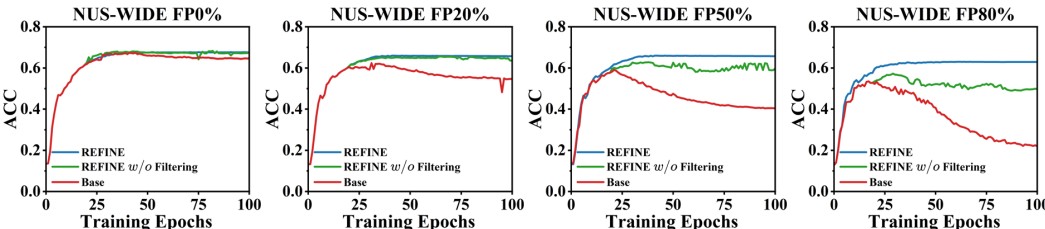

Figure 5: Clustering accuracy trends on the NUS-WIDE dataset under different FP ratios.

**Study of periodic cluster prototype initialization**: Figure 3 (b) shows the influence of prototype initialization intervals $e_f$ on model performance across the Caltech-101 and NUS-WIDE datasets under both 20% and 50% FP ratios. We observe that $e_f = 20$ (*i.e.*, initializing the cluster prototypes every 20 epochs) yields the best performance. In contrast, $e_f = \infty$ (*i.e.*, initializing only once after warmup) achieves the worst result, confirming the importance of periodic prototype refinement. While $e_f = 1$ (*i.e.*, initializing every epoch) achieves performance comparable to the optimal setting (*i.e.*, $e_f = 20$), it incurs substantially greater computational overhead. Considering the trade-off between performance and efficiency, $e_f = 20$ offers the most effective balance.

**Sensitivity analysis of loss weights**: Figure 3 (c) illustrates the sensitivity analysis of loss weight parameters on NUS-WIDE with FP ratios of 20% and 50%. We vary the consistency loss weight $\gamma_1$ and entropy regularization weight $\gamma_2$ within $\{0.1, 1, 10, 100\}$. Results show that REFINE achieves the best performance when $\gamma_1 = 10$ and $\gamma_2 = 1$. Notably, our method exhibits strong robustness to variations in $\gamma_2$, while the selection of $\gamma_1$ proves crucial for obtaining optimal clustering results.

**Study of threshold** $\eta$: Figure 3 (d) studies the impact of the singular threshold $\eta$ (varied in $\{0.0, 0.2, 0.4, 0.6, 0.8, 1.0\}$) on Caltech-101 and NUS-WIDE under 20% and 50% FP ratios. Our experimental results reveal two key findings: (1) REFINE maintains consistent performance across the entire range of $\eta \in [0, 0.8]$, demonstrating remarkable robustness to this hyperparameter, and (2) optimal clustering accuracy is achieved at $\eta = 0.4$, suggesting this value provides the ideal balance for feature space regularization.

**Visualization Analysis**: Figure 6 shows the T-SNE visualizations on the NUS-WIDE dataset, comparing our REFINE with CANDY (Guo et al., 2024) under 20% and 50% FP ratios. The plots depict $k$-means results on learned semantic features, colored by ground truth. As shown, our REFINE produces more compact and well-separated clusters than CANDY. This demonstrates our stronger robustness to unaligned samples.

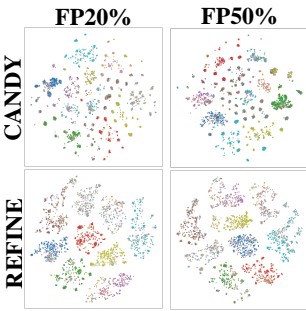

Figure 6: T-SNE visualization on the NUS-WIDE dataset.

## 5 CONCLUSION

This paper proposed a robust Partially View-aligned Clustering (PVC) approach REFINE, addressing the challenging scenario where sample alignment is not strictly required during training. We developed a Cross-view Semantics-based Filtering strategy to systematically identify and filter out cross-view mismatches. This filtering mechanism was applied during both prototype initialization and cross-view consistency learning, enabling the model to focus on semantically reliable samples and thereby improving clustering robustness. Moreover, we introduced a Shared-space Contrastive Learning framework that conducted contrastive learning in a unified latent space to reduce modality gaps. Extensive experiments on diverse benchmark datasets have validated REFINE's effectiveness.

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

# A APPENDIX

## A.1 CODE

Our code is anonymously available at `https://github.com/REFINE-REFINE/REFINE`.

## A.2 ALGORITHM

Algorithm 1 demonstrates the training procedure of our proposed framework. During the warmup stage, the model learns initial view-invariant representations by computing intra-view and inter-view hard contrastive losses. After warmup, consistent sample filtering is performed every fixed number of epochs $e_f$ to initialize the cluster prototypes for each view. In each iteration, the cluster prototypes are used to further select reliable samples for computing the KL loss and entropy regularization. Meanwhile, embeddings from all views are generated through a shared cross-view decoder to compute the soft contrastive loss. Finally, the total loss is used to update the network parameters.

---

**Algorithm 1** Algorithm of REFINE

---

Input: Training data $D = \{(x_i^{(v)}, x_i^{(u)})\}_{i=1}^N$, warmup epoch $E_{\text{warm}}$, max epoch $E_{\text{max}}$, cluster prototype initialization interval $e_f$.
**for** $epoch = 1, 2, \ldots, E_{\text{warm}}$ **do**
    **for** $iteration = 1, 2, \ldots$ **do**
        Calculate $\mathcal{L}_{\text{intra}}$ and $\mathcal{L}_{\text{inter}}$ by Eq. (3)
        Update network $\theta$ by optimizing $\mathcal{L}_{\text{warm}}$ by Eq. (5)
    **end for**
**end for**
**for** $epoch = E_{\text{warm}}, \ldots, E_{\text{max}}$ **do**
    **if** $(epoch - E_{\text{warm}}) \bmod e_f = 0$ **then**
        Select reliable samples $I_{\text{align}}$ using $D$ by Eq. (7)
        Get prototypes $C_{\text{final}}^{(v)}, C_{\text{final}}^{(u)}$ using $I_{\text{align}}$ by Eq. (8)
    **end if**
    **for** $iteration = 1, 2, \ldots$ **do**
        Select reliable samples $I_{\text{trusted}}$ by Eq. (10)
        Calculate $\mathcal{L}_{\text{cont}}$ using $I_{\text{trusted}}$ by Eq. (11)
        Calculate $\mathcal{L}_{\text{ent}}$ using $I_{\text{trusted}}$ by Eq. (13)
        Calculate $\mathcal{L}_{\text{intra}}^{\text{soft}}$ and $\mathcal{L}_{\text{inter}}^{\text{soft}}$ by Eq. (18)
        Update network $\theta$ by optimizing $\mathcal{L}_{\text{total}}$ by Eq. (19)
    **end for**
**end for**
Output: Updated network $\theta$

---

## A.3 COMPLEXITY ANALYSIS

We analyze the time complexity of our REFINE after the warmup stage. The total training complexity is dominated by three components: periodic cluster prototype initialization, cross-modal semantic consistency learning, and shared-space contrastive learning. For clarity, we denote the number of views as $V$, number of clusters as $K$, feature dimension as $d$, dataset size as $N$, number of $k$-means iterations as $T$, total training epochs as $E_{\text{max}}$, warmup epochs as $E_{\text{warm}}$, cluster initialization interval as $e_f$, and the number of aligned samples as $|I_{\text{align}}|$.

**Periodic Cluster Prototype Initialization**: This module involves multi-view clustering, semantic alignment, and consistent sample selection. The computational cost consists of:

- Multi-view $k$-means clustering: $O(VTNKd)$;
- Hungarian matching across views: $O(VK^3)$;
- Consistent sample selection across views: $O(VN)$;
- Re-clustering over aligned samples: $O(VT|I_{\text{align}}|Kd)$;

- Re-matching after re-clustering: $O(VK^3)$.

Thus, the total cost of a single initialization is $O(VTNKd + VK^3 + VN)$. Since this module is executed every $e_f$ epochs after warmup, the total complexity over training is: $O\left(\frac{E_{\max} - E_{\text{warm}}}{e_f} \cdot (VTNKd + VK^3 + VN)\right)$

**Cross-modal Semantic Consistency Learning**: This module includes pseudo-label computation, trusted sample selection, KL loss, and entropy regularization:

- Soft pseudo-labels computation via Student's t-distribution: $O(VNKd)$;
- Trusted sample selection across views: $O(V^2N)$;
- Symmetric KL divergence loss: $O(V^2NK)$;
- Entropy regularization: $O(VNK)$.

Above all, the total complexity is: $\mathcal{O}((E_{\max} - E_{\text{warm}}) \cdot (VNKd + V^2NK))$

**Shared-space Contrastive Learning**: This part comprises affinity matrices computation, high-order graph construction, SVD-based denoising, and contrastive loss:

- Affinity matrices computation: $O(V^2Nd)$;
- High-order graph construction: $O(V^2N)$;
- SVD-based denoising: $O(V^2N)$;
- Contrastive loss: $O(V^2N)$.

Above all, the total training complexity is: $O\left((E_{\max} - E_{\text{warm}}) \cdot V^2N\right)$

## A.4 EXPERIMENTS

**Datasets Details:** We conduct experiments on five benchmark datasets, namely Scene15 (Fei-Fei & Perona, 2005), Caltech101 (Li et al., 2015), LandUse-21 (Li et al., 2015), Reuters (Amini et al., 2009), and NUS-WIDE (Hu et al., 2019). The detailed characteristics are illustrated in Table 4.

Table 4: Statistics of the benchmark datasets used in our experiments.

| Dataset | Views | Type | Dimension | Samples | Classes |
|---|---|---|---|---|---|
| Scene15 | 2 | Image | 20/59 | 4485 | 15 |
| Caltech-101 | 2 | Image | 4096/4096 | 8677 | 101 |
| LandUse21 | 2 | Image | 59/40 | 2100 | 21 |
| Reuters | 2 | Text | 10/10 | 18758 | 6 |
| NUS-WIDE | 2 | Image | 4096/300 | 9000 | 10 |

**Network Architecture Details:** REFINE consists of three learnable modules: view-specific Siamese encoder, shared cross-view decoder, and view-specific cluster layer. The details of the network architectures are presented in Table 5.

**Evaluation Metrics:** We report four widely used clustering metrics: Accuracy (ACC), Normalized Mutual Information (NMI), Adjusted Rand Index (ARI), and Fowlkes-Mallows Score (FMS). All metrics range from 0 to 1, with higher values indicating better clustering quality.

- **Clustering Accuracy (ACC)**:

$$\text{ACC} = \frac{1}{N} \sum_{i=1}^{N} \mathbf{1}\{y_i = \text{map}(\hat{y}_i)\}, \tag{20}$$

where $N$ is the number of samples, $y_i$ is the ground-truth label of the $i$-th sample, $\hat{y}_i$ is the predicted cluster, and $\text{map}(\cdot)$ denotes the permutation function that provides the best one-to-one mapping between predicted clusters and ground-truth labels.

Table 5: Architecture of view-specific encoder, cross-view decoder, and cluster layer. "dim$^{(v)}$" denotes the input dimension of the $v$-th view, and "C" denotes the number of cluster centroids.

| Dataset | Encoder | Decoder | Cluster Layer (C $\times$ d) |
|---|---|---|---|
| Scene-15 | Linear(dim$^{(v)}$, 1024), BatchNorm, ReLU, Dropout
Linear(1024, 1024), BatchNorm, ReLU, Dropout
Linear(1024, 1024), BatchNorm, ReLU, Dropout
Linear(1024, 128), BatchNorm | Linear(128, 512), ReLU
Linear(512, 128) | 15 $\times$ 128 |
| Caltech101 | Linear(dim$^{(v)}$, 1024), BatchNorm, ReLU, Dropout
Linear(1024, 1024), BatchNorm, ReLU, Dropout
Linear(1024, 1024), BatchNorm, ReLU, Dropout
Linear(1024, 128), BatchNorm | Linear(128, 512), ReLU
Linear(512, 128) | 101 $\times$ 128 |
| LandUse21 | Linear(dim$^{(v)}$, 1024), BatchNorm, ReLU, Dropout
Linear(1024, 1024), BatchNorm, ReLU, Dropout
Linear(1024, 1024), BatchNorm, ReLU, Dropout
Linear(1024, 128), BatchNorm | Linear(128, 512), ReLU
Linear(512, 128) | 21 $\times$ 128 |
| NUS-WIDE | Linear(dim$^{(v)}$, 1024), BatchNorm, ReLU, Dropout
Linear(1024, 1024), BatchNorm, ReLU, Dropout
Linear(1024, 1024), BatchNorm, ReLU, Dropout
Linear(1024, 128), BatchNorm | Linear(128, 512), ReLU
Linear(512, 128) | 10 $\times$ 128 |
| Reuters | Linear(dim$^{(v)}$, 128), BatchNorm, ReLU, Dropout
Linear(128, 1024), BatchNorm, ReLU, Dropout
Linear(1024, 1024), BatchNorm, ReLU, Dropout
Linear(1024, 64), BatchNorm | Linear(64, 256), ReLU
Linear(256, 64) | 6 $\times$ 64 |

- **Normalized Mutual Information (NMI)**:

$$\text{NMI}(Y, \hat{Y}) = \frac{2I(Y; \hat{Y})}{H(Y) + H(\hat{Y})}, \tag{21}$$

where $Y$ and $\hat{Y}$ are the sets of ground-truth and predicted cluster assignments, respectively, $I(A; B)$ presents the mutual information, and $H(\cdot)$ denotes Shannon entropy.

- **Adjusted Rand Index (ARI)**:

$$\text{ARI} = \frac{\text{RI} - \mathbb{E}[\text{RI}]}{\max(\text{RI}) - \mathbb{E}[\text{RI}]}, \tag{22}$$

where RI is the Rand Index that measures pairwise label agreement, $\mathbb{E}[\text{RI}]$ is its expected value under random assignment, and $\max(\text{RI})$ is the maximum possible RI.

- **Fowlkes-Mallows Score (FMS)**:

$$\text{FMS} = \sqrt{\frac{TP}{TP + FP} \cdot \frac{TP}{TP + FN}}, \tag{23}$$

$TP$, $FP$, and $FN$ denote true positives, false positives, and false negatives, respectively.

**Experimental Results:** To provide a more comprehensive evaluation, we report the mean and standard deviation of each metric over five independent runs of our method REFINE, as shown in Table 6. This statistical reporting highlights the stability and consistency of our method under varying FP ratios. Notably, the experiments under higher FP ratios tend to exhibit slightly larger standard deviations (*i.e.*, most exceeding 1.0), which may be attributed to the increased impact of misaligned samples during the warm-up phase.

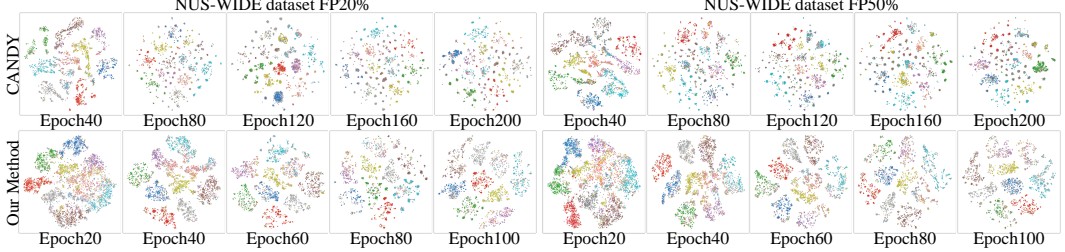

Figure 7: T-SNE visualization on NUS-WIDE dataset with FP ratios of 20% and 50%.

Table 6: Results on five datasets, reported as the mean±standard deviation over 5 independent runs.

| FP Ratio | Scene15 | Caltech-101 | LandUse21 | Reuters | NUS-WIDE |
|---|---|---|---|---|---|
| | ACC / NMI / ARI / FMS (%) | | | | |
| 0% | 44.4±0.70 | 68.3±0.82 | 32.5±0.99 | 52.1±3.40 | 67.0±1.72 |
| | 46.0±0.59 | 84.8±0.33 | 39.0±0.75 | 34.1±3.03 | 54.3±1.13 |
| | 26.1±0.41 | 72.9±0.58 | 18.1±0.30 | 28.5±5.08 | 49.5±1.60 |
| | 30.5±0.92 | 73.7±0.64 | 20.1±0.46 | 42.7±1.36 | 54.2±1.65 |
| 20% | 41.7±0.98 | 66.7±0.51 | 31.9±0.54 | 55.9±3.60 | 66.9±0.29 |
| | 43.1±0.74 | 83.8±0.26 | 38.1±0.37 | 32.5±2.54 | 52.3±0.40 |
| | 24.0±0.80 | 72.3±0.62 | 17.5±0.26 | 33.9±3.78 | 48.2±0.42 |
| | 29.5±0.49 | 73.0±0.71 | 19.9±0.25 | 44.88±1.21 | 53.1±0.57 |
| 50% | 40.8±0.19 | 63.4±0.62 | 31.5±0.74 | 55.0±4.31 | 64.1±0.60 |
| | 41.4±0.18 | 82.1±0.54 | 37.3±0.45 | 30.8±1.87 | 48.3±0.86 |
| | 23.4±0.32 | 69.0±0.44 | 16.8±0.29 | 28.7±2.86 | 44.2±0.72 |
| | 28.7±0.23 | 71.3±0.90 | 19.6±0.39 | 44.4±1.70 | 44.6±0.27 |
| 80% | 40.8±1.24 | 54.6±1.24 | 30.1±1.41 | 39.1±4.45 | 57.2±0.96 |
| | 39.7±1.02 | 76.5±1.31 | 34.5±1.75 | 15.5±4.80 | 41.8±0.62 |
| | 23.2±1.12 | 64.6±0.50 | 15.5±1.29 | 13.6±4.43 | 36.9±0.70 |
| | 27.9±0.82 | 65.2±2.41 | 19.2±0.78 | 32.1±1.54 | 44.3±0.65 |

**T-SNE Visualization Analysis:** In Figure 7, we present T-SNE visualizations on NUS-WIDE to compare the clustering performance of our REFINE with the SOTA method CANDY (Guo et al., 2024) under 20% and 50% FP ratios. The plots show $k$-means clustering results of the learned semantic features, with points colored by ground truth labels. For fair comparison, we adopt CANDY's official code and settings. As shown, REFINE produces more compact and well-separated clusters with only 100 epochs, while CANDY, even at 200 epochs, yields less distinct clusters. This indicates that REFINE better resists the noise from unaligned samples and more effectively captures the underlying cluster structure, resulting in improved semantic consistency and robustness in PVC.

**Convergence Analysis:** We analyze the convergence behavior of REFINE by tracking its performance metrics (ACC, NMI, and ARI) over training epochs on NUS-WIDE dataset, under 20% and 50% FP ratios. As shown in Figure 8, the performance steadily improves during the early training stages and gradually stabilizes after around 100 epochs, indicating that the model has effectively converged. This trend holds under both noise settings, demonstrating the robustness of REFINE in handling partially unaligned samples.

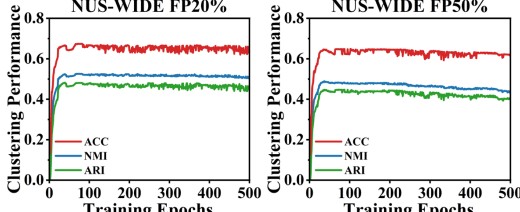

Figure 8: Clustering performance of our REFINE with increasing epoches on NUS-WIDE dataset with FP ratios of 20% and 50%.

## A.5 LIMITATIONS AND FUTURE WORK

Although our REFINE demonstrates strong performance in PVC settings, it has some limitations. Since prototype updating is sensitive to unstable samples, misclassifications by the sample filtering mechanism may perturb the learned cluster representations, leading to suboptimal results. In the future, we will focus on refining the filtering strategy and enhancing prototype updating to improve robustness against noisy or ambiguous samples and strengthen overall clustering stability.

## A.6 THE USE OF LARGE LANGUAGE MODELS (LLMS)

Large language models (LLMs) were employed exclusively for minor language polishing. They were not used for ideation, data analysis, or other substantive aspects of this work.

