# OpenReview forum: "Robust Semantic Sample Filtering for Partially View-aligned Clustering"
_ICLR.cc/2026/Conference — Submitted to ICLR 2026_

### Official Review · Reviewer_71jf · 2025-10-21

**Soundness:** 3
**Presentation:** 4
**Contribution:** 2
**Rating:** 4
**Confidence:** 4

**Summary:**

The paper proposes REFINE, a new  multi-view clustering (MVC) method. REFINE focuses on the problem of partial view alignment in MVC. Previous studies often assume that the correspondence between views is known and accurate. In contrast, this paper introduces a new and meaningful scenario, where it is unknown whether the inter-view correspondence is available or not. To address this challenge, the authors observe that the model tends to first learn clearer cross-view correspondences and then gradually adapt to more complex cross-view relationships in partially aligned data. Based on this insight, they design an effective solution. Extensive experiments demonstrate the effectiveness of the proposed method. Compared with previous works, the proposed method achieves remarkably good performance in most scenarios.

**Strengths:**

1. The motivation of this work is relevant and meaningful, as in real-world scenarios it is often challenging to determine whether the correspondences across multi-view datasets are accurate or contaminated by noise. From this perspective, the paper makes a valuable addition.

2. The proposed method is conceptually simple yet effective. It assesses the correctness of inter-view correspondences by measuring the consistency among clustering assignments across views. Extensive experiments validate the robustness and effectiveness of this approach.

3. The paper is clearly written and easy to read. Moreover, the experimental evaluation is relatively thorough and provides solid support for the proposed method.

**Weaknesses:**

1. The compared baselines in the experiments are somewhat dated. It would strengthen the paper to include comparisons with more recent related methods. Although there may not be existing works that address exactly the same setting as REFINE, such comparisons would still be informative and valuable, particularly when the FP Ratio is 0.

2. The authors mention that the proposed method is inspired by the sample selection strategy used in NLL. It would be beneficial to provide a brief overview or discussion of this prior strategy in the related work section to better contextualize the contribution.

3. An obvious observation is that when the FP Ratio becomes large, the performance of all methods tends to decrease to some extent. This raises a concern: under such conditions, could the performance of MVC methods be inferior to clustering based on a single view?  For example, suppose the correspondence between two views (View 1 and View 2) is incorrect, and REFINE produces clustering result $a$, while performing clustering on the two views independently yields results $b$ and $c$. If $a$ < min($b$, $c$), it may suggest that the incorrect inter-view consistency leads to severe degradation, to the extent that the benefit of multi-view learning is largely lost. From this perspective, it might be helpful to include single-view clustering results as additional baselines to provide a more comprehensive comparison and better illustrate the effect of misaligned correspondences.

**Questions:**

1. As the proposed approach may be the first to address the case where inter-view correspondences are unknown, it would be beneficial for the authors to clarify the experimental settings in the comparative studies. Specifically, were the compared methods evaluated under the assumption that inter-view correspondences are known? This clarification would help readers better understand the fairness and validity of the comparisons.

2. From a methodological perspective, Section 3.1 emphasizes the use of Siamese encoders. In the scenario considered by REFINE, what are the advantages of using Siamese encoders compared with the more commonly used autoencoders?

---

> ### Author Response · Authors · 2025-11-19
> **Response to reviewer 71jf**
>
> Thank you for reviewing our work.
>
> $\textbf{Q1 More Comparisons:}$ We have added comparisons with recent methods in the revised version (Table 2), including RMCNC [A] (2024), DDMVC [B] (2025), and MCMVC [C] (2025). All results are reported directly from their respective papers.
>
>
> $\textbf{Q2 Sample Selection in NLL:}$ We have revised the related work to clarify our sample filtering motivation (2.2). Existing noisy label learning (NLL) approaches can be broadly classified into three main categories: loss-based methods, sample-based methods, and label-based methods. Among these, sample-based NLL methods are our focus. Sample-based methods often rely on a selection strategy that identifies a subset of samples with high-confidence predictions, which are then used to guide training while avoiding the influence of noisy labels. For example, [D] selects clean samples using a Jensen-Shannon (JS) divergence metric. [E] distinguishes training samples using a dual-module Gaussian Mixture Model (GMM). Drawing inspiration from sample selection strategies in NLL, REFINE uses cross-view semantic consistency as the primary criterion: pairs with inconsistent predicted cluster assignments are identified as potentially misaligned samples and filtered during training.
>
> $\textbf{Q3 Single View Performance:}$ We added single-view clustering results. Experimental results show that REFINE maintains competitive performance on several datasets (e.g., Scene15, Caltech101, NUS-WIDE) even under extreme 80% misalignment vs single-view, demonstrating its ability to preserve multi-view advantages under severe noise. While single-view may slightly outperform in some highly misaligned cases, we emphasize the core value of multi-view clustering lies in leveraging the potential cross-modal complementarity, rather than simply replacing single-view performance. REFINE’s filtering of inconsistent samples provides a foundation for future improvements, such as realignment or consistency reconstruction.
>
> |Dataset|Single-View (ACC/NMI/ARI)|REFINE (ACC/NMI/ARI) under FP 80%|
> |--|--|--|
> |Scene15|36.6/20.9/36.6|40.8/39.7/23.2|
> |Caltech101|54.5/76.8/37.4|54.6/76.5/64.6|
> |LandUse21|32.5/38.2/17.7|30.1/34.5/15.5|
> |Reuters|39.8/20.6/15.8|39.1/15.5/13.6|
> |NUS-WIDE|48.3/37.6/27.7|57.2/41.8/36.9|
>
> $\textbf{Q4 Experimental Settings:}$ CANDY is the first work addressing multi-view clustering under unknown inter-view correspondences. In our experiments, we strictly followed CANDY’s data processing and evaluation protocols and directly adopted their reported results. This ensures that all comparisons are conducted under the same unknown-correspondence setting, guaranteeing fairness and consistency.
>
> $\textbf{Q5 Siamese Encoder:}$ Following [F] [G], we adopt a Siamese encoder based on the following considerations: (1) Representation alignment: The Siamese encoder independently generates latent representations for each view. These representations are then aligned in a shared latent space via a shared decoder and cross-view losses (e.g., KL divergence loss and contrastive loss), mitigating semantic bias caused by view misalignment. In contrast, traditional autoencoders focus on reconstruction and may preserve view-specific noise. (2) Efficiency: The Siamese architecture is more lightweight than a dual-branch autoencoder, with fewer parameters, making it suitable for large-scale or multi-view scenarios while preserving both feature expressiveness and cross-view alignment.
>
>
> [A] Robust Multi-View Clustering with Noisy Correspondence
>
> [B] Deep multi-view clustering with diverse and discriminative feature learning
>
> [C] Explicit View-Labels Matter: A Multifacet Complementarity Study of Multi-View Clustering
>
> [D] Jo-SRC: A Contrastive Approach for Combating Noisy Labels
>
> [E] DivideMix: Learning with Noisy Labels as Semi-supervised Learning
>
> [F] Robust Contrastive Multi-view Clustering against Dual Noisy Correspondence
>
> [G] Decoupled Contrastive Multi-view Clustering with High-order Random Walks

---

> > ### Comment · Reviewer_71jf · 2025-11-26
> >
> > Thank you for your response. Some of my concerns have been addressed, so I have decided to raise my score.

---

> > > ### Author Response · Authors · 2025-11-26
> > > **Response to reviewer 71jf**
> > >
> > > Thank you so much for your careful review and recognition! We will keep moving to make our REFINE better and better.

---

### Official Review · Reviewer_w53J · 2025-10-27

**Soundness:** 3
**Presentation:** 3
**Contribution:** 2
**Rating:** 4
**Confidence:** 4

**Summary:**

This paper introduces REFINE to tackle multi-view clustering with noisy correspondence. In the CSF module, it filters out possibly misaligned samples to improve the quality of cluster centers, and enforces cross-modal semantic consistency with symmetric KL divergence minimization. In the SCL module, a shared decoder for different modalities to reduce the modality gap. Combining these components, REFINE outperforms pervious SOTA methods, achieving better semantic consistency and better separated distribution. Extensive comparison and ablation results confirm the effectiveness of the proposed method.

**Strengths:**

1. By employing a shared cross-view decoder to project different modalities into the same latent space, REFINE not only reduces modality gaps but also strengthens the effectiveness of the filtering mechanism.

2. REFINE consistently outperforms state-of-the-art methods across five benchmark datasets and various noise levels (false positive ratio up to 80%), surpassing recent methods such as CANDY and DIVIDE.

3. The paper is well written, and the proposed method is validated with a wide range of ablation studies and visualizations.

4. The paper studies an practical problem for the multi-view clustering community.

**Weaknesses:**

1. Although the paper achieves better performance, the improvements are relatively marginal (<2%) in the multi-view clustering community. Moreover, there is no direct evaluation of filtering module (e.g., precision/recall).

2. The paper claims that the shared decoder reduces modality gaps, but does not provide comparative experiments with independent decoders.

3. Minor writing and formatting issues, including improper usage of `\citet` / `\citep`.

4. The overall novelty is limited. The network architecture is very resemble to the recent SOTA methods like CANDY.

5. The experiment evaluations are not solid enough. Concretely, according to Table 1, the hyper-parameters $L_r$ and $\gamma_{1}$ vary among different datasets, which might violate the unsupervised characteristic of multi-view clustering. Moreover, the effectiveness of the method is only verified on small scaled datasets (all less than 20,000).

**Questions:**

1. Beyond final clustering metrics, could you please report filtering precision/recall (against known ground truth correspondences) to directly validate the quality of the selected samples?

2. Why shared decoder is better than independent decoders per view, especially when the embedding spaces differ substantially?

3. Since the paper discusses extending to more than two views, could you comment on the performance trends as the number of views increases?

Please refer to the weaknesses to see more questions.

---

> ### Author Response · Authors · 2025-11-19
> **Response to reviewer w53J**
>
> Thank you for reviewing our work.
>
> $\textbf{Q1 Marginal Performance Gains:}$ Our method tackles an under-explored Partially View-aligned Clustering (PVC) setting: unknown partial view misalignment. By leveraging semantic consistency to filter misaligned samples, REFINE reduces interference from forced alignment and enables future use of these samples (e.g., re-matching, restoration, etc.). Importantly, in high-noise settings (50% and 80%), REFINE achieves improvements over SOTA CANDY (the first to explore this problem), with average ACC ↑2-3.5%, NMI ↑2.4-4.7%, and ARI ↑3.7-4.9%, demonstrating robustness under severe misalignment.
>
> $\textbf{Q2 Filtering Precision/Recall:}$ We have added analysis of filtering precision in revised version (Figure 4). Specifically, $I_\text{aligned}$ is the initial filtered set (Eq. (7)), directly affected by misaligned samples. $I_\text{trusted}$ is derived from the prediction probabilities of the prototype classifier (Eq. (10))—whose prototypes are initialized via k-means on $I_\text{aligned}$—thereby further reducing noise. Results show that REFINE effectively removes a large portion of misaligned samples while maintaining high precision for $I_{\text{trusted}}$. And even under 50% FP, where $I_\text{aligned}$ precision slightly drops due to severe misalignment, $I_\text{trusted}$ remains relatively robust ($I_{\text{aligned}}$→$I_{\text{trusted}}$ ↑27.81-24.25%). These findings confirm that our semantics-based filtering successfully transfers NLL sample selection to the PVC setting.
> |Dataset|FP Ratio|$I_{\text{aligned}}$ Precision (First→Last)|$I_{\text{trusted}}$ Precision (First→Last)|
> |-|-|-|-|-|
> |NUS-WIDE|20%|82.53 → 82.22|95.61 → 94.90|
> |NUS-WIDE|50%|55.81 → 54.90|83.62 → 79.35|
>
> $\textbf{Q3 Decoder Experiment:}$ We conducted an ablation study on NUS-WIDE by using independent decoders. As shown, the shared decoder consistently outperforms independent decoders. Theoretically, a shared cross-view decoder aligns embeddings into a common latent space, reducing modality gaps and improving semantic consistency across views. In contrast, independent decoders may cause embeddings to drift apart, making cross-view alignment more difficult, particularly when the embedding distributions differ significantly.
> |Dataset|FP Ratio|Independent Decoders (ACC/ARI/NMI)|Shared Decoder (Ours) (ACC/ARI/NMI)|
> |-|-|-|-|
> |NUS-WIDE|0%|65.1/51.6/45.4|67.0/54.3/49.5|
> |NUS-WIDE|20%|63.9/49.9/44.2|66.9/52.3/48.2|
> |NUS-WIDE|50%|61.8/47.0/42.6|64.1/48.3/44.2|
> |NUS-WIDE|80%|54.4/38.7/33.8|57.2/41.8/36.9|
>
> $\textbf{Q4 Writing Issues:}$ We have corrected the relevant writing in the revised version.
>
> $\textbf{Q5 Novelty vs CANDY:}$ While our architecture is based on CANDY, the main novelty of REFINE lies in introducing sample selection strategy in NLL for PVC. Unlike CANDY, which focuses on robust loss design and may underfit correct correspondences, we directly address misalignment. The NLL-inspired filtering strategy enables the identification of misaligned samples, helping model fully learn from correctly aligned samples and laying the foundation for future work on leveraging misaligned ones.
>
> $\textbf{Q6 Hyperparameters:}$ We adopt different hyper-parameters based on a coarse grid search combined with k-fold cross-validation, as different datasets exhibit varying learning difficulty.
>
> $\textbf{Q7 Larger Datasets and More Views}$: To evaluate performance trends with increasing numbers of views and on larger datasets, we conducted experiments on CIFAR-100 (50k samples, 3 views) and Caltech101 (2–5 views) following [A][B]. The results show that REFINE consistently maintains or improves clustering performance as the number of views increases and performs well on moderately sized datasets, demonstrating its robustness and scalability.
>
> |Dataset|Method|ACC|NMI|
> |-|-|-|-|
> |CIFAR-100|MFLVC|82.68|95.60|
> |CIFAR-100|GCFAgg|95.97|99.35|
> |CIFAR-100|REFINE (Ours)|99.97|99.96|
>
> |FP Ratio|Method|Caltech-2V (ACC/NMI)|Caltech-3V (ACC/NMI)|Caltech-4V (ACC/NMI)|Caltech-5V (ACC/NMI)|
> |-|-|-|-|-|-|
> |0%|WFLVC|60.60/52.80|63.10/56.60|73.30/65.20|80.40/70.30|
> |0%|GCFAgg|66.43/50.08|64.00/53.45|73.43/66.10|83.36/73.31|
> |0%|REFINE (Ours)|69.79/49.42|75.93/67.78|80.64/71.36|85.21/75.15|
> |50%|MFLVC|47.36/28.17|54.43/52.83|61.36/54.83|60.79/41.75|
> |50%|GCFAgg|35.57/20.64|33.79/15.68|29.57/10.58|28.57/9.57|
> |50%|REFINE (Ours)|58.93/40.03|64.29/49.04|68.14/51.73|71.29/54.09|
>
>
> [A] GCFAgg: Global and Cross-view Feature Aggregation for Multi-view Clustering
>
> [B] Multi-level feature learning for contrastive multi-view clustering.

---

> ### Author Response · Authors · 2025-11-27
> **Response to reviewer w53J**
>
> Dear Reviewer,
>
> I hope this message finds you well. As the discussion period is approaching its end with under a week remaining, I would like to confirm that we have fully addressed your concerns. If there are any additional points or feedback you'd like us to consider, please let us know. Your insights are invaluable to us, and we are eager to address any remaining issues to improve our work.
>
> Thank you for your time and effort in reviewing our paper.

---

### Official Review · Reviewer_LVPN · 2025-10-27

**Soundness:** 3
**Presentation:** 2
**Contribution:** 2
**Rating:** 6
**Confidence:** 4

**Summary:**

This paper tackles the partially view-aligned clustering problem, where cross-view correspondences between samples are incomplete or noisy. The authors propose a framework integrating two core modules: cross-view semantics-based filtering and shared-space contrastive learning. Experiments on five datasets demonstrate good performance under both fully aligned and highly misaligned conditions.

**Strengths:**

1.The proposed method demonstrates robust performance even under severe misalignment (up to 80%), showing good robust.

2.Extensive ablation and sensitivity analyses clearly validate the contribution of each module to the overall performance.

3.The shared latent space design effectively enhances cross-view alignment by reducing view gaps and improving feature consistency.

**Weaknesses:**

1.Over-reliance on pseudo-label consistency. The filtering mechanism assumes pseudo-labels from K-means are reliable indicators of semantic correctness, which may not hold early in training. Misleading pseudo-labels could generate filtering errors.

2.Discarding instead of reusing uncertain samples. All inconsistent samples are dropped, reducing data utilization under high misalignment.

3.The problem of scalability to more views. Although the paper mentions that the proposed method can be naturally scaled to more than two views, the datasets chosen for the experiment still consist of only two views.

**Questions:**

1.How does the filtering mechanism behave when early pseudo-labels are highly noisy?

2.Can the proposed method handle more extreme cases (e.g., 90% misalignment)?

---

> ### Author Response · Authors · 2025-11-19
> **Response to reviewer LVPN**
>
> Thank you for your thoughtful review and positive feedback.
>
> $\textbf{Q1 Over-Reliance on Pseudo-Label Consistency:}$ We acknowledge this risk. Inspired by NLL learning dynamics, we assume the model first fits clean samples before noisy ones. During warm-up, we use hard pairwise alignment to accelerate initial learning, making subsequent filtering of misaligned samples more robust. Empirically, we evaluated this on NUS-WIDE under various noise levels across training epochs (Figure 5, revised version). Here, ''Base'' trains using only warm-up losses. Its performance shows a rise-then-fall trend, indicating that despite misleading pseudo-labels, the model initially fits correct samples before overfitting noisy ones. This aligns with our assumption, supporting the robustness of our filtering mechanism.
>
> $\textbf{Q2 Discarding instead of Reusing:}$ We acknowledge that discarding inconsistent or uncertain samples reduces data utilization to some extent. Our key idea, inspired by NLL sample selection, transfers this concept to multi-view clustering to identify misaligned samples, laying the foundation for future work on correcting or better leveraging them. Discarding uncertain samples is a practical trade-off: it slightly reduces data usage but ensures reliable cross-view semantic consistency learning, especially under high misalignment, and helps improve training stability and performance.
>
>
> $\textbf{Q3 Scalability to More Views:}$ Following GCFAgg [A] and MFLVC [B], we conducted 2–5 view experiments on Caltech. The results show that REFINE naturally and effectively scales to multi-view settings, and demonstrates stronger robustness when views are misaligned.
> |FP Ratio|Method|Caltech-2V (ACC/NMI)|Caltech-3V (ACC/NMI)|Caltech-4V (ACC/NMI)|Caltech-5V (ACC/NMI)|
> |----|----|----|----|----|----|
> |0%|WFLVC|60.60/52.80|63.10/56.60|73.30/65.20|80.40/70.30|
> |0%|GCFAgg|66.43/50.08|64.00/53.45|73.43/66.10|83.36/73.31|
> |0%|REFINE (Ours)|69.79/49.42|75.93/67.78|80.64/71.36|85.21/75.15|
> |50%|MFLVC|47.36/28.17|54.43/52.83|61.36/54.83|60.79/41.75|
> |50%|GCFAgg|35.57/20.64|33.79/15.68|29.57/10.58|28.57/9.57|
> |50%| REFINE (Ours)|58.93/40.03|64.29/49.04|68.14/51.73|71.29/54.09|
>
> $\textbf{Q4 Filtering Behavior under Extreme Noise:}$ To further evaluate REFINE under severe misalignment, we conducted clustering experiments with a 90% FP ratio and tracked both clustering performance (ACC/NMI/ARI) and filtering performance measured by $I_{\text{trusted}}$ F1 (from initial to final epochs). Results confirm that the filtering mechanism remains effective on most datasets ($I_{\text{trusted}}$ F1 increases as training progresses), maintaining reasonable clustering performance even under such extreme noise levels.
> |Dataset|ACC|NMI|ARI|$I_{\text{trusted}}$ F1 (First→Last)|
> |----|----|----|----|----|
> |Caltech101|49.51|70.67|57.52|46.76 → 58.33|
> |LandUse21|28.33|31.63|13.63|16.67 → 26.30|
> |NUS-WIDE|49.01|35.37|29.74|43.03 → 47.42|
> |Reuters|36.10|12.78|11.67|44.86 → 41.55|
> |Scene15|38.46|37.13|20.90|24.30 → 39.19|
>
>
> [A] GCFAgg: Global and Cross-view Feature Aggregation for Multi-view Clustering
>
> [B] Multi-level feature learning for contrastive multi-view clustering

---

### Official Review · Reviewer_5DEh · 2025-10-28

**Soundness:** 3
**Presentation:** 3
**Contribution:** 2
**Rating:** 4
**Confidence:** 4

**Summary:**

This paper proposes REFINE, a framework for Partially View-Aligned Clustering, which combines Cross-view Semantics-based Filtering and Shared-space Contrastive Learning to improve robustness under view misalignment. It is inspired by Noisy Label Learning, which filter unreliable sample correspondences via semantic consistency. The paper is technically solid and experimentally comprehensive, but suffers from some weaknesses in clarity, novelty positioning, and mathematical justification.

**Strengths:**

REFINE tackles the practically pertinent issue of view misalignment in multi-view clustering with clear motivation, integrates the semantically based filtering and shared-space contrastive learning modules in an intuitively complementary fashion. The authors conduct comprehensive experiments across five benchmark datasets under varying false-positive ratios, supported by  ablation studies that firmly attest to the contribution of each component.

**Weaknesses:**

1.The FP simulation is implemented via random shuffling. It is recommended to supplement the evaluation with a real-world misalignment dataset to verify the robustness of REFINE under naturally occurring cross-view inconsistencies.
2.When all views are severely inconsistent, will the Semantics-based Filtering module discard too many samples, thereby reducing the effective training data and harming clustering stability? Is there any theoretical bound or empirical estimation on the filtering precision (i.e., the false discard rate of reliable samples) to ensure that the semantic filtering mechanism does not eliminate too many valid correspondences?
3.The entropy regularization term ( L_{\text{ent}} ) is designed to prevent cluster collapse and encourage a uniform marginal distribution across clusters. However, in real-world datasets, the true class distribution is often imbalanced. Would this term force the model to learn an overly uniform clustering structure that does not reflect the underlying data distribution?

**Questions:**

1.How does REFINE perform when only one view provides strong, informative features while other views are weak or noisy? Have related experiments been conducted to verify robustness under such conditions?
2.in Eq. 9, why is the Student’s t-distribution preferred over the standard softmax function for pseudo-label assignment? Please clarify the theoretical or empirical rationale behind this design.

---

> ### Author Response · Authors · 2025-11-19
> **Response to reviewer 5DEh**
>
> Thank you for taking time to review.
>
> $\textbf{Q1 Real World Dataset:}$ While real-world multi-view datasets with structured misalignment are limited, we conducted additional experiments on Noisy MNIST following COMPLETER [A], using 20k MNIST images as view 1 and within-class images with Gaussian noise as view 2, providing preliminary evidence of robustness under weak or noisy cross-view conditions.
> |Dataset|Method|ACC|NMI|ARI|
> |-|-|-|-|-|
> |Noisy MNIST|COMPLETER|89.08|88.86|85.47|
> |Noisy MNIST|REFINE (Ours)|87.50|84.22|80.88|
>
> $\textbf{Q2 Filtering Robustness and Theoretical Bound:}$
> Our method is built on the intuition that models fit clean samples before memorizing noise. By using cross-modal semantic consistency, we identify reliable sample correspondences.
> Experiments below show this mechanism effectively balances noise reduction and data preservation.
>
> (1) In the updated version Figure 5, we compared three settings on NUS-WIDE under different FP ratios: the full method (REFINE), a variant without filtering (REFINE $w/o$ Filtering), and a baseline using only warmup loss (Base). Results show: "Base" severely overfits under high noise; "REFINE $w/o$ Filtering" gains from semantic learning but remains unstable; "REFINE" improves robustness and clustering stability under noise through adaptive filtering.
>
> (2) We further evaluated REFINE under 90% FP ratio and tracked clustering performance along with filtering precision ($I_\text{trusted}$ F1). Results indicate that even under extreme noise, the filtering mechanism dynamically preserves sufficient reliable samples to maintain reasonable clustering performance.
> |Dataset|ACC|NMI|ARI|$I_\text{trusted}$ F1 (First→Last)|
> |----|----|----|----|----|
> |Caltech101|49.51|70.67|57.52|46.76 → 58.33|
> |LandUse21|28.33|31.63|13.63|16.67 → 26.30|
> |NUS-WIDE|49.01|35.37|29.74|43.03 → 47.42|
> |Reuters|36.10|12.78|11.67|44.86 → 41.55|
> |Scene15|38.46|37.13|20.90|24.30 → 39.19|
>
> While theoretical bound is beyond the scope of this work, as it would require additional assumptions such as noise patterns, we do not claim universal guarantees. Instead, we provide strong evidence via empirical results to support the effectiveness of our approach.
>
> $\textbf{Q3 Entropy Regularization Term:}$ We acknowledge that real-world datasets may have imbalanced class distributions, and the entropy regularization encourages a more uniform marginal cluster distribution. In our method, this regularization is applied only to the marginal distribution of trusted samples, rather than all samples, reducing the risk of forcing uniformity across the entire dataset. Moreover, similar entropy-based regularizations are widely used in multi-view clustering [B][C], which have been shown to improve clustering stability and performance.
>
> $\textbf{Q4 Student’s t-distribution:}$ We use the Student’s t-distribution in Eq. (9) for pseudo-label assignment because it provides a heavier tail than the standard softmax, which helps emphasize close neighbors while reducing the influence of distant points. This property is particularly useful in clustering and representation learning, as seen in prior work [D][E]. Empirically, we compared using softmax versus the Student’s t-distribution on NUS-WIDE. These results confirm that the Student’s t-distribution improves clustering performance by providing sharper, more discriminative pseudo-label assignments compared to softmax.
> |Dataset|FP Ratio|Softmax (ACC/ARI/NMI)|Student’s t-distribution (ACC/ARI/NMI)|
> |----|----|----|----|
> |NUS-WIDE|0%|65.42/51.60/46.84|67.0/54.3/49.5|
> |NUS-WIDE|20%|64.18/49.34/45.07|66.9/52.3/48.2|
> |NUS-WIDE|50%|61.31/44.84/40.42|64.1/48.3/44.2|
> |NUS-WIDE|80%|49.02/33.19/27.08|57.2/41.8/36.9|
>
> [A] COMPLETER: Incomplete Multi-view Clustering via Contrastive Prediction
>
> [B] Image Clustering with External Guidance
>
> [C] Deep Multiview Clustering by Contrasting Cluster Assignments
>
> [D] Unsupervised Deep Embedding for Clustering Analysis
>
> [E] Deep Embedded Multi-View Clustering via Jointly Learning Latent Representations and Graphs

---

> ### Author Response · Authors · 2025-11-27
> **Response to reviewer 5DEh**
>
> Dear Reviewer,
>
> I hope this message finds you well. As the discussion period is approaching its end with under a week remaining, I would like to confirm that we have fully addressed your concerns. If there are any additional points or feedback you'd like us to consider, please let us know. Your insights are invaluable to us, and we are eager to address any remaining issues to improve our work.
>
> Thank you for your time and effort in reviewing our paper.

---

### Official Review · Reviewer_GBMU · 2025-11-01

**Soundness:** 2
**Presentation:** 2
**Contribution:** 2
**Rating:** 2
**Confidence:** 4

**Summary:**

This paper tackles *partially view‑aligned clustering* (PVC), where cross‑view correspondences are unknown and potentially noisy. The proposed framework, REFINE, combines (i) Cross‑view Semantics‑based Filtering—used both for Periodic Cluster Prototype Initialization (PCPI) and for *Cross‑modal Semantic Consistency Learning* (CSCL)—and (ii) a Shared‑space  Contrastive Learning* (SCL) module with spectral denoising. Architecturally, each view is encoded by Siamese encoders (online/key), query features are projected through a shared cross‑view decoder into a unified space, and view‑specific cluster heads produce soft assignments; Figure 2 (page 3) gives a clear overview and the filtering submodule on the right shows how consistent pairs are retained for prototype re‑initialization and KL‑based consistency training.

**Strengths:**

Using semantic agreement to *filter* pairs is intuitive and well‑motivated by noisy label learning. The two insertion points—prototype init and consistency learning—are complementary, and the shared decoder plausibly reduces cross‑view modality gaps (Figure 2, page 3).

   The evaluation spans five benchmarks and four FP settings with repeated runs. The ablation suite is unusually careful for clustering work (Table 3 and Figure 3), and the periodic re‑init analysis (Figure 3b) is useful to practitioners.

   Entropy regularization against collapse, SVD‑based spectral denoising to soften supervision, and the use of Hungarian matching for class‑level alignment are appropriate for this setup (Sec. 3, pages 5–7).

**Weaknesses:**

Warm‑up assumes hard pairwise alignment in the loss. (Sec. 3.1, )
Claims of consistent” SOTA are not fully supported by Table 2.
 Evaluation mostly on *synthetic* misalignment; limited real‑world evidence. For eg. All PVC settings are constructed by random shuffling with a fixed FP ratio (Sec. 4.1, page 7). That is useful but may not capture structured misalignment from acquisition lags, occlusions, or domain shifts. Demonstrations on genuinely misaligned multi‑modal datasets (e.g., image–text collections with partial captions or desynchronized multi‑sensor logs) would strengthen the “real‑world robustness” claim.

Under‑specified architecture and complexity details.

Risk of confirmation bias in filtering.

Although Table 1 lists core hyperparameters (page 7), several training choices are as in CANDY, and no standard deviations are reported. Given small absolute margins in some settings, error bars and a short note on per‑baseline tuning would increase credibility.

**Questions:**

Can you ablate *disabling inter‑view contrastive* during warm‑up (i.e., only intra‑view) or replace identity‑based positives with mined high‑agreement pairs to quantify the cost/benefit of early hard alignment noise? (Sec. 3.1, Eq. (3))

---

> ### Author Response · Authors · 2025-11-19
> **Response to reviewer GBMU**
>
> Thank you for taking time to review.
>
> $\textbf{Q1 Hard Alignment during Warm-up:}$ Following Noisy Label Learning (NLL) dynamics, we apply hard inter-view alignment in warm-up to first fit clean pairs and accelerate convergence, which later helps filtering noisy ones. To quantify its effect, we disable inter-view alignment during warm-up (only intra-view). On NUS-WIDE with different FP, "Only Intra-View" clear drops (ACC ↓3–5%, NMI ↓1–5%, ARI ↓2–7%,). This confirms that early cross-view alignment is beneficial despite noisy pairs.
> |Dataset|FP ratio|Only Intra-View (ACC/NMI/ARI)|Full Method (ACC/NMI/ARI)|
> |-|-|-|-|
> |NUS-WIDE|0%|64.1/50.9/46.4|67.0/54.3/49.5|
> |NUS-WIDE|20%|61.1/47.1/41.8|66.9/52.3/48.2|
> |NUS-WIDE|50%|59.1/44.9/38.9|64.1/48.3/44.2|
> |NUS-WIDE|80%|53.2/40.3/34.6|57.2/41.8/36.9|
>
> $\textbf{Q2 Consistent SOTA:}$ We acknowledge that our wording was imprecise and have revised it (4.3). While REFINE does not always achieve absolute SOTA in every scenario, it consistently improves performance on average. Specifically, under high FP ratios (50% and 80%), ACC ↑2-3.5%, NMI ↑2.4-4.7%, and ARI ↑3.7-4.9% on average, which we consider substantial improvements.
>
> $\textbf{Q3 Real World Robustness:}$ While real-world multi-view datasets with structured misalignment are limited, we conducted additional experiments on Noisy MNIST following COMPLETER[A], using 20k MNIST images as view 1 and within-class images with Gaussian noise as view 2, providing preliminary evidence of robustness under cross-view inconsistencies.
> |Dataset|Method|ACC|NMI|ARI|
> |-|-|-|-|-|
> |Noisy MNIST|COMPLETER|89.08|88.86|85.47|
> |Noisy MNIST|REFINE (Ours)|87.50|84.22|80.88|
>
> $\textbf{Q4 Architecture Clarification:}$ The framework includes three learnable modules: a view-specific Siamese encoder, a shared cross-view decoder, and a view-specific cluster layer. Both online and target encoders are fully connected networks (FCNs), with each layer sequentially followed by BatchNorm, ReLU, and Dropout, following [B]. The final BatchNorm layer in the encoder does not have learnable affine parameters. The cross-view decoder is a two-layer MLP and the cluster layer contains learnable centroids. We have updated the manuscript (4.2 & A.4 Table5).
>
> $\textbf{Q5 Complexity Details:}$ For clarity, we denote the number of views as $V$, number of clusters as $K$, feature dimension as $d$, dataset size as $N$, number of $k$-means iterations as $T$, total training epochs as $E_\text{max}$, warmup epochs as $E_\text{warm}$, and cluster initialization interval as $e_f$. Overall, after the warmup stage, the total complexity is dominated by periodic clustering $O\Big(\frac{E_\text{max}-E_\text{warm}}{e_f}(V T N K d + V K^3 + V N)\Big)$, cross-modal consistency learning $O\big((E_\text{max}-E_\text{warm})(V N K d + V^2 N K)\big)$, and contrastive learning $O\big((E_\text{max}-E_\text{warm}) V^2 N\big)$. More details are in the revised version (A.3).
>
> $\textbf{Q6 Bias Risk:}$ We acknowledge the potential risk of confirmation bias in filtering. To mitigate this, we adopt: (1) Periodic cluster prototype initialization, allowing previously filtered samples to re-enter selection. (2) Prototype refinement based on initially filtered samples before selecting trusted samples, which minimizes the influence of misaligned data.
>
> $\textbf{Q7 Hyperparameters and Standard Deviations:}$ We adopt dataset-specific learning rates and epochs based on a coarse grid search combined with k-fold cross-validation, as different datasets exhibit varying learning difficulty. Furthermore, we have added the mean and standard deviation over 5 runs for REFINE to improve the reliability of the results in the revised version (A.4 Table 6).
> |FP Ratio|Scene15 (ACC/NMI/ARI/FMS)|Caltech-101 (ACC/NMI/ARI/FMS)|LandUse21 (ACC/NMI/ARI/FMS)|Reuters (ACC/NMI/ARI/FMS)|NUS-WIDE (ACC/NMI/ARI/FMS)|
> |-|-|-|-|-|-|
> |0%|44.4±0.70/46.0±0.59/26.1±0.41/30.5±0.92|68.3±0.82/84.8±0.33/72.9±0.58/73.7±0.64|32.5±0.99/39.0±0.75/18.1±0.30/20.1±0.46|52.1±3.40/34.1±3.03/28.5±5.08/42.7±1.36|67.0±1.72/54.3±1.13/49.5±1.60/54.2±1.65|
> |20%|41.7±0.98/43.1±0.74/24.0±0.80/29.5±0.49|66.7±0.51/83.8±0.26/72.3±0.62/73.0±0.71|31.9±0.54/38.1±0.37/17.5±0.26/19.9±0.25|55.9±3.60/32.5±2.54/33.9±3.78/44.9±1.21|66.9±0.29/52.3±0.40/48.2±0.42/53.1±0.57|
> |50%|40.8±0.19/41.4±0.18/23.4±0.32/28.7±0.23|63.4±0.62/82.1±0.54/69.0±0.44/71.3±0.90|31.5±0.74/37.3±0.45/16.8±0.29/19.6±0.39|55.0±4.31/30.8±1.87/28.7±2.86/44.4±1.70|64.1±0.60/48.3±0.86/44.2±0.72/44.6±0.27|
> |80%|40.8±1.24/39.7±1.02/23.2±1.12/27.9±0.82|54.6±1.24/76.5±1.31/64.6±0.50/65.2±2.41|30.1±1.41/34.5±1.75/15.5±1.29/19.2±0.78|39.1±4.45/15.5±4.80/13.6±4.43/32.1±1.54|57.2±0.96/41.8±0.62/36.9±0.70/44.3±0.65|
>
> [A] COMPLETER: Incomplete Multi-view Clustering via Contrastive Prediction
>
> [B] Robust Contrastive Multi-view Clustering against Dual Noisy Correspondence

---

> ### Author Response · Authors · 2025-11-27
> **Response to reviewer GBMU**
>
> Dear Reviewer,
>
> I hope this message finds you well. As the discussion period is approaching its end with under a week remaining,I would like to confirm that we have fully addressed your concerns. If there are any additional points or feedback you'd like us to consider, please let us know. Your insights are invaluable to us, and we are eager to address any remaining issues to improve our work.
>
> Thank you for your time and effort in reviewing our paper.

---

### Author Response · Authors · 2025-11-19
**Response to all reviewers**

We sincerely thank the reviewers for their constructive feedback. In response, we have substantially revised the manuscript to address the concerns raised. Key improvements are summarized as follows:

$\textbf{1. Sample selection related work}$

$\textbf{2. Detailed model architectures}$

$\textbf{3. Comparisons with more SOTA methods}$

$\textbf{4. Filtering precision analysis}$

$\textbf{5. Robustness analysis of NLL-inspired filtering}$

$\textbf{6. Writing issues}$


Due to space constraints, several deeper analyses and extended experiments were included in the response and supplementary material but not in the main manuscript. These include:

$\textbf{1. Experiments on larger datasets and more views scenarios}$ (i.e., Noisy MNIST, CIFAR-100, Caltech 2-5 views).

$\textbf{2. Statistical analysis}$ (i.e., experimental statistics including mean and standard deviation over 5 runs)

$\textbf{3. Additional ablation studies}$ (i.e., hard alignment vs inter-view alignment during warm-up, student’s distribution vs softmax, independent decoder vs shared decoder, single view vs multi-view)

$\textbf{4. Robustness analysis under extreme noise rates}$ (i.e., clustering performance and filtering F1 under FP 90% ).

$\textbf{5. Detailed computational complexity analysis}$


Please see our detailed responses below each review, and all the changes in the revised paper are high-lighted with blue font. We appreciate your feedback and welcome any additional questions.

---

### Author Response · Authors · 2025-12-01
**Summary Comment for the Area Chair and Reviewers**

We would like to thank Reviewer LVPN (initial score 6) and Reviewer 71jf (raised the score from 4 to 6) for their positive evaluation of our work. For the remaining reviewers, we regret that they had not responded before the discussion period ended, and we are no longer able to continue the exchange.

During the discussion phase, we addressed comments in detail and revised the paper accordingly (the main changes are summarized in our previous response).

We sincerely thank all reviewers and the AC for their time and efforts throughout the review process.

---

### Meta-Review · Area_Chair_vGEr · 2026-01-05

**Summary:**

Reviewer GBMU: Using semantic agreement to filter pairs is intuitive and well‑motivated by noisy label learning. The evaluation spans five benchmarks and four FP settings with repeated runs. Entropy regularization against collapse, SVD‑based spectral denoising to soften supervision, and the use of Hungarian matching for class‑level alignment are appropriate for this setup. However, the reviewer still has some concerns on the weaknesses about the hard pairwise alignment assumption in the loss, under‑specified architecture and complexity details, Risk of confirmation bias in filtering.

 Reviewer 5DEh: The method tackles the practically pertinent issue of view misalignment in multi-view clustering with clear motivation, integrates the semantically based filtering and shared-space contrastive learning modules in an intuitively complementary fashion. The authors conduct comprehensive experiments across five benchmark datasets under varying false-positive ratios.However, the reviewer still has some concerns on the weaknesses about the lack of more evaluation with a real-world misalignment dataset, lack of theoretical bound or empirical estimation on the filtering precision.

Reviewer LVPN: The proposed method demonstrates robust performance even under severe misalignment (up to 80%). Extensive ablation and sensitivity analyses clearly validate the contribution of each module to the overall performance. However, the reviewer still has some concerns on the weaknesses about the over-reliance on pseudo-label consistency, discarding instead of reusing uncertain samples, scalability problems to more views.

 Reviewer w53J: The method not only reduces modality gaps but also strengthens the effectiveness of the filtering mechanism. REFINE consistently outperforms state-of-the-art methods across five benchmark datasets. The paper is well written, and the proposed method is validated with a wide range of ablation studies and visualizations. However, the reviewer still has some concerns on the weaknesses about the  relatively marginal improvements, lack of more experiments, writing issues, limited novelty, not solid experiment evaluations.

Reviewer 71jf: The motivation of this work is relevant and meaningful. The proposed method is conceptually simple yet effective. The paper is clearly written and easy to read. However, the reviewer still has some concerns on the weaknesses about the dated compared baselines, lack of more related works, lack of more experimental evaluations.

**Reviewer Concerns:**

After carefully evaluating the rebuttals, I think the reviews from the Reviewer LVPN abd 71jf were addressed from the response.
For the remaining reviewer concerns, they are all not fully addressed.

**Reviewer Scores:**

For the Reviewer Reviewer GBMU, 5DEh, and w53J, I think the reviewer may keep the rating unchanged based on the response.

For the  Reviewer  LVPN abd 71jf, I think the reviewer may increase the rating or keep the rating unchanged based on the response.

---

### Decision · Program_Chairs · 2026-01-26

Reject